# RETHINKING OOD DETECTION AT SCALE THROUGH ENSEMBLE DIVERSITY

## ABSTRACT

The common practice of equating in-distribution (ID) data with the training set is a flawed oversimplification for large-scale applications. A viable strategy for out-of-distribution (OOD) detection is to train an ensemble of models to disagree, a method proven effective at smaller scales. However, these approaches have been limited by their reliance on external OOD datasets for diversification. This work revisits the fundamental definition of OOD data based on data density. This perspective reveals that the low-density, "OOD-like" samples required for diversification are already present within large training sets, removing the need for external data. We introduce the loss-guided diversification regulariser (LDR) to operationalise this principle. LDR identifies these internal samples by targeting those with high cross-entropy loss and encourages the ensemble to disagree only on them, thereby learning diverse, generalisable hypotheses. To ensure scalability, LDR employs a stochastic pairing strategy which reduces computational complexity from quadratic to constant. The process also yields a new uncertainty metric, the predictive diversity score (PDS). Extensive evaluation on benchmarks like ImageNet shows that LDR, combined with PDS, achieves state-of-the-art performance in OOD detection. Our work demonstrates that sourcing disagreement from within the training set is a powerful and effective paradigm for building robust models at scale.

## 1 INTRODUCTION

Detecting out-of-distribution samples is critical for robust and reliable AI systems. Historically, the boundary between in-distribution (ID) and out-of-distribution (OOD) data has been the training set itself (Hendrycks & Gimpel, 2017; Liang et al., 2018; Hendrycks et al., 2019; Hein et al., 2019; Lee et al., 2018; Du et al., 2022; Sun et al., 2022; Huang & Li, 2021). Samples from the training distribution were considered ID; everything else was OOD. This binary view is a flawed oversimplification. It fails to capture the reality of modern, web-scale machine learning. Large datasets such as ImageNet are not clean, monolithic distributions. They are heterogeneous collections which contain not only "prototypical" ID examples but also a long tail of rare instances, ambiguous cases, and functionally OOD samples like cartoons (Swayamdipta et al., 2020; Northcutt et al., 2021; Beyer et al., 2020; Hendrycks et al., 2021c; Liu et al., 2019; Sambasivan et al., 2021). In this work, we re-adopt the original definition based on **data density** (Liu et al., 2020; Lee et al., 2018; Sun et al., 2022; Van Amersfoort et al., 2020; Charpentier & Günnemann, 2020). Under this definition, large training sets inevitably contain OOD samples, even if in small quantities.

**Ensemble disagreement** is a viable proxy for low data density. The intuition is that a diverse set of models will agree on high-density data where evidence is plentiful. They will diverge in their predictions on low-density data where they must extrapolate. Prior work has explored this connection and has trained diverse ensembles that use disagreement as a signal for OOD detection (Pagliardini et al., 2023; Lee et al., 2023; Teney et al., 2022a;b; Benoit et al., 2024). These approaches, however, were limited to smaller-scale problems. They relied on a crucial and often impractical assumption; the availability of a separate, external OOD dataset. This dataset served as the explicit training ground for disagreement. This requirement has limited a powerful idea to small-scale settings and prevented its application to the massive datasets where OOD detection is most critical (Hendrycks et al., 2022; Hendrycks & Gimpel, 2017; Huang & Li, 2021; Hendrycks et al., 2019). The reliance on external data stems from viewing the training set as a monolithic ID entity. This view breaks down at scale

(Swayamdipta et al., 2020; Northcutt et al., 2021; Beyer et al., 2020; Hendrycks et al., 2021c; Liu et al., 2019; Sambasivan et al., 2021). This density-based perspective reveals that the necessary low-density, "OOD-like" samples are already present within large training sets (Liu et al., 2020; Charpentier & Günnemann, 2020). While the scale of modern datasets makes OOD detection more challenging (Hendrycks et al., 2022; Huang & Li, 2021; Yang et al., 2024), they also contain the necessary ingredients for a scalable, ensemble-based solution.

Our **loss-guided diversification regulariser (LDR)** leverages the availability of these internal OOD samples. LDR sources disagreement samples from within a large-scale training set, using high cross-entropy loss as its guide. Forcing disagreement on these internal samples encourages the ensemble to learn multiple, diverse mechanisms and hypotheses. Each mechanism provides a low-loss explanation for the core ID data. This diversity causes the ensemble to disagree when faced with novel ambiguity. To operate at the ImageNet scale, LDR employs practical strategies like stochastic sums, which reduce computational complexity from quadratic to constant. This process also yields a new uncertainty metric, the **predictive diversity score (PDS)**, which quantifies the degree of ensemble disagreement.

We demonstrate the effectiveness of our approach through extensive empirical validation. Our method significantly increases ensemble diversity across multiple benchmarks compared to existing methods. When combined, LDR and PDS achieve state-of-the-art OOD detection performance on datasets including ImageNet-C, iNaturalist, and OpenImages, outperforming prior diversification techniques and other strong baselines. Furthermore, the diversification induced by LDR directly translates to improved OOD generalisation accuracy. These results prove that sourcing disagreement from within the training set is a powerful and effective paradigm for scalable diversification.

## 2 RELATED WORK

In this section, we review prior work on the diverse ensembles application to OOD detection. Several studies have shown that ensemble disagreement provides a useful signal for OOD detection (Pagliardini et al., 2023; Lee et al., 2023; Tifrea et al., 2022; Lakshminarayanan et al., 2017; Everett et al., 2022; Linmans et al., 2020; Yashima et al., 2022; Trinh et al., 2024; Zaidi et al., 2021), since disagreement between ensemble components reflects epistemic uncertainty that tends to be higher on OOD samples than on ID samples (Mukhoti et al., 2023). This goes in line with the results on uncertainty estimation benchmarks that also highlight the efficacy of disagreement-based scores for OOD detection (Mucsányi et al., 2024; de Jong et al., 2024). Yet, no prior work has demonstrated how to train ensembles on ImageNet-scale data such that their disagreement aligns with the theoretical definition of OOD. Existing methods either (i) do not provide any guarantees of disagreement being a good proxy for OOD (Tifrea et al., 2022; Lakshminarayanan et al., 2017; Everett et al., 2022; Linmans et al., 2020; Yashima et al., 2022; Trinh et al., 2024; Scimeca et al., 2023), or (ii) require an auxiliary OOD dataset to induce disagreement (Pagliardini et al., 2023; Lee et al., 2023; Teney et al., 2022a;b). Thus, at ImageNet scale, the first are ineffective (Xia & Bouganis, 2022) and the second infeasible to train due to blurred ID – OOD boundaries. In this paper, we show how to train diverse ensembles that overcome both limitations. See § A.1 for details on techniques to induce ensemble disagreement.

## 3 PROPOSED METHOD

We revisit the definition of ID as high-density regions of the input space and OOD as low-density ones. On small scales, this reduces to treating the training set as ID and everything else as OOD. At a large scale, however, this equivalence does not necessarily hold. To sidestep this limitation, in § 3.1, we refer to an alternative separation of ID and OOD via diverse ensembles, where low density is captured by high output diversity among members.

This perspective has been explored so far only on a small scale, using external OOD samples for diversification. We introduce the preliminary materials on existing approaches to diversification in §3.2. We identify two challenges in scaling up ensemble diversification: (1) where to source the OOD data and (2) how to overcome the quadratic complexity $O(M^2)$ of pairwise diversification terms for $M$ models in an ensemble.

We notice that on a larger scale, one can diversify the models based on the training set itself, as a large-scale training set will contain OOD samples. We address (1) in §3.3 by proposing Loss-guided Diversification Regularizer (LDR). We address (2) in §A.5.1 by proposing to apply the diversification term on stochastic pairs in the Stochastic Gradient Descent (SGD) iterations. Afterwards, we introduce our novel measure of epistemic uncertainty, Predictive Diversity Score (PDS), in §3.4.

**What is ID and OOD?** We start with a definition that distinguishes ID and OOD at the level of the underlying distributions:

**Definition 1** (Distributional definition of OOD). *Let $\mathcal{X}$ be an input space and $\mathcal{Y}$ be a label space, $\varepsilon > 0$ some small threshold and $\text{supp}_\varepsilon(P) = \{(x,y) \in \mathcal{X} \times \mathcal{Y} : P(x,y) > \varepsilon\}$ (we fix one $\varepsilon$ throughought the paper and call $\text{supp}_\varepsilon$ as supp for brevity). Then we say that $(x,y)$ is ID sample $(x \sim P)$ iff $(x,y) \in \text{supp}(P)$, and OOD sample $(x \sim Q)$ iff $(x,y) \in \text{supp}(Q)$, where $\text{supp}(Q) = (\mathcal{X} \times \mathcal{Y}) \setminus \text{supp}(P)$.*

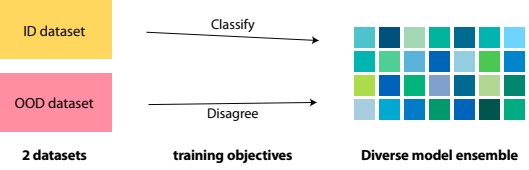

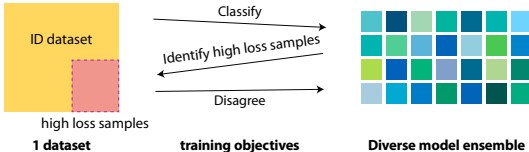

Figure 1: **Existing diversification methods (top)** require distinct (unlabeled) OOD examples on which the models are encouraged to disagree. Our **Loss-guided Diversification Regularizer (LDR, bottom)** instead encourages the models to diverge on high loss examples identified within a single standard training set.

A popular way to define in-distribution (ID) and out-of-distribution (OOD) data is to say that ID is everything contained in the training set, and OOD is everything else (Hendrycks & Gimpel, 2017; Liang et al., 2018; Hendrycks et al., 2019; Hein et al., 2019; Lee et al., 2018; Du et al., 2022; Sun et al., 2022; Huang & Li, 2021). However, this view fails when the training set itself includes out-of-distribution points, which often occurs due to annotation errors or noisy web-sourced pipelines.

**Setting.** Our training data is $\mathcal{D} := \{x_n, y_n\}_{n=1}^N \sim \text{Mix}(P, Q)$, it is sampled from a mixture of in-distribution (ID) data from distribution $P$ with minor addition of out-of-distribution (OOD) data from distribution $Q$. Prior diversification methods based on "prediction disagreement" Lee et al. (2023); Pagliardini et al. (2023) require a separate set of unlabeled OOD examples $\mathcal{D}^{\text{ood}} := \{x_n^{\text{ood}}\}_{n=1}^{N^{\text{ood}}} \sim Q$. Our proposed method will show how to proceed without $\mathcal{D}^{\text{ood}}$. We denote with $f(\cdot, \theta) \in \mathcal{F}$ a neural network classifier of parameters $\theta$ from hypothesis class $\mathcal{F} \subseteq \mathcal{Y}^\mathcal{X}$. Then $f(x; \theta) \in \mathbb{R}^C$ corresponds to the logits over $C$ classes for the input $x$, and $p(x) := \text{Softmax}(f(x)) = e^{f(x)}/\sum_{c=1}^C e^{f_c(x)} \in [0,1]^C$ probabilities over the classes. Our goal is to obtain an ensemble $\{f^1, \cdots, f^M\}$ of $M$ models. Our experiments in Section 4 will showcase multiple ways to exploit these models.

## 3.1 DISAGREEMENT AS A PROXY FOR OOD

The distribution-based definition of ID and OOD is difficult to quantify in practice, since the original distributions are unknown. A practical alternative is to reason not directly with training samples, but with their solutions – functions that achieve zero loss on data drawn from $P$. Formally:

**Theorem 1** (Disagreement-based characterization of OOD). *Use notation from Definition 1. For an ensemble of solutions*

$$\{f^1, \ldots, f^M\} \subseteq \{f \in \mathcal{F} : \Pr_{(x,y)\sim P}[f(x) \neq y] = 0\}, M > 1$$

*Define disagreement indicator: $J_M(x) := \mathbf{1}\{\exists m \neq m' : f^m(x) \neq f^{m'}(x), 1 \leq m, m' \leq M\}$. Under Assumptions 1–2:*

$$
\begin{aligned}
&\textit{(ID)} \qquad x \sim P \iff J_M(x) = 0, \\
&\textit{(OOD)} \qquad x \sim Q \iff J_M(x) = 1,
\end{aligned}
$$

*Thus, disagreement is an exact characterization of out-of-distribution inputs. See proof in Appendix A.3.*

This idealistic perspective also fails in the face of training distribution being sampled from a mixture of $P$ (ID) and $Q$ (OOD), because achieving $\Pr_{(x,y)\sim P}[f(x) \neq y] = 0$ in that case is impossible. We can therefore only approximate the ideal scenario by (i) minimizing the loss of the ensemble on $\mathcal{P}$, encouraging agreement on in-distribution samples, and (ii) maximizing disagreement on $\mathcal{Q}$, encouraging epistemic uncertainty on out-of-distribution samples. In the next section, we show how this approach can be implemented when separate OOD data are available for training disagreement.

## 3.2 Diversification through Disagreement

We introduce existing approaches to diversify an ensemble in scenarios when OOD data $\mathcal{D}^{\mathrm{ood}}$ is available. They encourage disagreement in predictions using the main objective of solving the original task (e.g. image classification) while enforcing an auxiliary training objective (Lee et al., 2023; Pagliardini et al., 2023) that rewards different predictions on the OOD split $\mathcal{D}^{\mathrm{ood}}$. For a set of models $\{f^m\}_{m=1}^{m=M}$, the main training objective is given by the cross-entropy loss over all $M$ ensemble members and $N$ training examples:

$$\mathcal{L}_{\mathrm{main}} := \frac{1}{MN} \sum_n^N \sum_m^M - \log p_{y_n}^m (x_n; \theta). \tag{1}$$

The auxiliary disagreement objective is applied to every pair of models and every OOD example in $\mathcal{D}^{\mathrm{ood}}$:

$$\mathcal{L}_{\mathrm{div}} := \frac{1}{N^{\mathrm{ood}}M(M-1)} \cdot \sum_{n=1}^{N^{\mathrm{ood}}} \sum_{m=1}^{M} \sum_{l=1}^{m-1} \mathcal{G}\big(p^m(x_n^{\mathrm{ood}}),\, p^l(x_n^{\mathrm{ood}})\big). \tag{2}$$

The $\mathcal{G}(\cdot, \cdot)$ term leads to diversification by encouraging a pair of models $(f^m, f^l)$ to make different predictions on samples from $\mathcal{D}^{\mathrm{ood}}$. Our method applies to several implementations of $\mathcal{G}$ from the existing literature (D'Angelo & Fortuin, 2021; Lee et al., 2023; Pagliardini et al., 2023). In our experiments, $\mathcal{G}$ implements the A2D ("Agree to disagree") objective from (Pagliardini et al., 2023):

$$\mathcal{G}\big(p^m(x),\, p^l(x)\big) := -\log\Big( p_{\hat{y}}^m(x) \cdot \big(1 - p_{\hat{y}}^l(x)\big) + p_{\hat{y}}^l(x) \cdot \big(1 - p_{\hat{y}}^m(x)\big) \Big) \tag{3}$$

where $\hat{y} := \arg\max_c p_c^m(x)$ is the class predicted by the model $p^m$ (the definition could just as well use the prediction from $p^l$, which would make no practical difference (Pagliardini et al., 2023)). Minimizing (3) encourages $p^l$ to assign a lower likelihood to the class predicted by $p^m$ and vice versa.

We claim that at a large scale, with millions of non-synthetic data points, the distinction between ID and OOD becomes blurred. In that case, the notion of OOD data is no longer well-defined even for humans, making $\mathcal{D}^{\mathrm{ood}}$ impossible to obtain. In the next section, we show how to sidestep the requirement for $\mathcal{D}^{\mathrm{ood}}$ for ensemble diversification.

## 3.3 Dynamic Selection of High Loss Examples

For the scenarios when $\mathcal{D}^{\mathrm{ood}}$ is not available, we introduce the key ingredient of our **Loss-guided Diversification Regularizer** (**LDR**): dynamic selection of high loss examples in training data as a replacement for the OOD data source in previous ensemble diversification approaches. With no OOD data, it is difficult to apply disagreement methods since the main training objective encourages all models to fit the training examples, hence to agree on all available data. In practice, extra OOD data for disagreement, which should clearly differ the ID data, may not be readily available as in the smaller-scale experiments with synthetic datasets like CIFAR, Waterbirds, and CelebA considered in previous work. Considering the scale of ImageNet or similar as the training data, it is not clear how to define and obtain data that qualify as OOD or where the feature-label correlations clearly differ.

We propose to replace the OOD disagreement data with a set of high loss training examples identified dynamically during training. The models are then encouraged to disagree on these examples. The desiderata for these high loss samples are twofold: (a) discriminate samples where the ensemble members make mistakes and (b) trust the ensemble prediction for the high loss sample identification only when the ensemble is sufficiently trained. Theoretical justification fot using high loss samples for disagreement is summarized in Theorem 2.

**Theorem 2** (Lower bound on OOD disagreement). *Given an ensemble $\{f^1, \ldots, f^M\} \subseteq \mathcal{F}$. Let $H \subset \mathcal{D}$ be a set of high cross-entropy loss samples:*

$$\{\exists \tau > 0 : \sum_m^M -\log p_y^m(x) \geq \tau, \forall (x, y) \in H\}.$$

*Assume that $\exists \delta_H > 0$, s.t. an empirical disagreement of ensemble on $H$ is lowerbounded by it:*

$$\frac{1}{|H|} \sum_{x \in H} \frac{1}{M(M-1)} \sum_{\substack{m, m'=1 \\ m \neq m'}}^{M} \mathbf{1}\{f^m(x) \neq f^{m'}(x)\} \geq \delta_H.$$

*Then, under Assumptions 3-6, the following holds:*

$$\Pr_{x \sim Q}\left[\exists m \neq m' : f^m(x) \neq f^{m'}(x)\right] \geq C_1(\tau) \cdot \delta_H - C_2(\tau).$$

*Where $C_1(\tau) > 0, C_2(\tau) > 0$ are constants monotone in $\tau$. $Q$ is OOD distribution. See constant definitions and proof in Appendix A.4.4.*

Informally, disagreement on high loss training samples increases disagreement on OOD data.

We assign a sample-wise weight $\alpha_n$ to each training sample $(x_n, y_n) \in \mathcal{D}$:

$$\alpha_n := \frac{\mathrm{CE}(\frac{1}{M} \sum_m f^m(x_n), y_n)}{\left(\frac{1}{|B|} \sum_{b \in B} \mathrm{CE}(\frac{1}{M} \sum_m f^m(x_b), y_b)\right)^2} \tag{4}$$

where $\mathrm{CE}(\frac{1}{M} \sum_m f^m(x_n), y_n)$ is the cross-entropy loss on the logit-averaged prediction and $B$ is a mini-batch that contains the sample $(x_n, y_n)$. $\alpha_n$ is a weight proportional to the ensemble loss on the sample, which fulfills desideratum (a) mentioned above. The normalization then handles desideratum (b). To see this, consider the batch-wise weight:

$$\alpha_B := \frac{1}{|B|} \sum_{b \in B} \alpha_b = \frac{1}{\frac{1}{|B|} \sum_b \mathrm{CE}(\frac{1}{M} \sum_m f^m(x_b), y_b)}. \tag{5}$$

Now $\alpha_B$ is *inversely proportional* to the average cross-entropy loss of the ensemble on the mini-batch $B$. Thus, the overall level of $\alpha_n$ for $n \in B$ is lower for earlier iterations of the ensemble training, where the predictions from the models are not trustworthy yet.

We now use the sample-wise weights $\alpha_n$ to define the LDR training objective:

$$\mathcal{L}_{\mathrm{LDR}} := \mathcal{L}_{\mathrm{main}} + \frac{\lambda}{NM(M-1)} \cdot \sum_n \sum_{m < l} \mathrm{stopgrad}(\alpha_n) \cdot \mathcal{G}\big(p^m(x_n), p^l(x_n)\big). \tag{6}$$

where $\lambda > 0$ controls the strength of the diversification. The operator $\mathrm{stopgrad}(\cdot)$ outputs a copy of its argument that is treated as a constant during backpropagation. Compared to Equation 2, this formulation does not require OOD disagreement data. Instead, all training examples are treated as potential high loss samples to disagree on, and their difficulty is softly determined via $\alpha_n$.

**Justification for the adaptive weights.** To justify the adaptive nature of $\alpha_n$, let us examine the gradient of the total loss (Equation 6). Considering an ensemble of two models $m$ and $l$, we have the gradient of the loss on a sample $(x, y)$ w.r.t. the model $m$'s predicted probability for the ground truth class ($p_y^m(x)$) (see Appendix A.14 for details):

$$\nabla_{p_y^m(x)} \mathcal{L}_{\mathrm{LDR}}(x, y) = -\frac{1}{p_y^m(x)} + \frac{\alpha_n(2p_y^l(x) - 1)}{C(m, l, y, x)}, \tag{7}$$

where the denominator $C(m, l, y, x)$ is some non-negative function that is upper-bounded by $1$. The gradient consists of the two terms. The sign of the first one, which corresponds to the cross-entropy, is always negative. The sign of the second one, which corresponds to the disagreement objective,

depends on the value of $p_y^l(x)$. The fact that the term can have different signs can cause training instabilities if none of the terms will dominate (have much higher absolute value comparing to another term) the total gradient.

The only way to control for that and avoid such instabilities is to make $\alpha_n$ proportional to $p_y^m(x)^{-1}$: $\alpha_n = \gamma p_y^m(x)^{-1}$ for some $\gamma > 0$. In such case the dominance of the total gradient by the second term is ensured when:

$$\frac{|2p_y^l(x) - 1|}{C(m, l, y, x)} \geq |2p_y^l(x) - 1| \geq \gamma^{-1}. \tag{8}$$

As a result, the total gradient will be lower on the correctly predicted samples and higher on the samples on which an ensemble makes mistakes (i.e. high loss samples) while being dominated by the disagreement term for sufficiently high values of $\gamma$. This allows for ensemble diversification without harming the approximation of the training distribution. In practice, we make $\alpha_n$ proportional to $-\log p_y^m(x)$ in equation 4, as it exhibits better diversification than using $p_y^m(x)^{-1}$.

### 3.4 PREDICTIVE DIVERSITY SCORE FOR OOD DETECTION

We now describe how to use diverse ensembles for OOD detection (Helton et al., 2004). This is based on evaluating the epistemic uncertainty, which is the consequence of the lack of training data in a given regions of the input space (Mukhoti et al., 2023; Hüllermeier & Waegeman, 2021). In these OOD regions, the lack of supervision means that diverse models are likely to disagree in their predictions (Malinin et al., 2019; Lee et al., 2023; Pagliardini et al., 2023). We therefore propose to use the agreement rate across models on a given sample to estimate the epistemic uncertainty and its "OODness".

**BMA Baseline.** Given an ensemble of models, a simple baseline for OOD detection is to compute the predictive uncertainty of the Bayesian Model Averaging (BMA) by treating the ensemble members as samples of the posterior $p(\theta|\mathcal{D})$ (Lakshminarayanan et al., 2017; Wilson & Izmailov, 2020):

$$\eta_{\text{BMA}} := \max_c \frac{1}{M} \sum_m p_c^m(x). \tag{9}$$

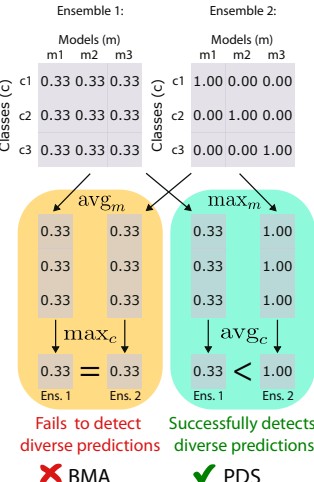

Figure 2: **BMA vs PDS**. BMA: Bayesian Model Averaging. PDS: Predictive Diversity Score. Given a sample $x$ and an ensemble of $M = 3$ models for $C = 3$ classes, which uncertainty scoring captures the ensemble diversity?

While being a strong baseline (Mukhoti et al., 2023) for OOD detection this notion of uncertainty does not directly exploit the potential diversity in individual models of the ensemble because it averages out the predictions along the model index $m$. In addition to that, mimicking the true distribution makes individual members have small values of $\max_c p_c^m(x)$ on training samples with high aleatoric uncertainty (Hüllermeier & Waegeman, 2021). This is why BMA is not a reliable indicator of epistemic uncertainty.

**Proposed Predictive Diversity Score (PDS).** We propose a novel measure for epistemic uncertainty that directly measures the prediction diversity of the individual members. Concretely,

$$\eta_{\text{PDS}} := \frac{1}{C} \sum_c \max_m p_c^m(x). \tag{10}$$

PDS is a continuous relaxation of the number of unique argmax predictions within an ensemble of models (#unique). To see this, consider the special case where $p^m \in \{0, 1\}$ are one-hot vectors. Then, $\max_m p_c^m(x)$ is 1 if any of $m$ predicts $c$ and 0 otherwise. Thus, in this example $\sum_c \max_m p_c^m(x)$ computes the number of classes predicted by at least one ensemble member. An illustrative case when PDS is preferable to BMA for epistemic uncertainty estimation can be seen in Figure 2.

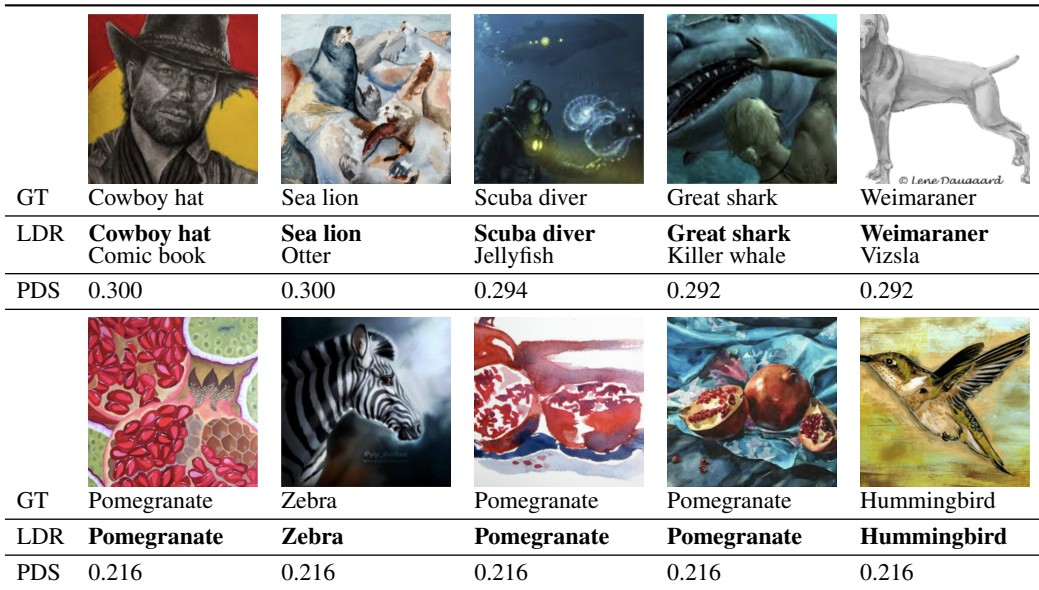

| | | | | | |
|---|---|---|---|---|---|
| GT | Cowboy hat | Sea lion | Scuba diver | Great shark | Weimaraner |
| LDR | **Cowboy hat** Comic book | **Sea lion** Otter | **Scuba diver** Jellyfish | **Great shark** Killer whale | **Weimaraner** Vizsla |
| PDS | 0.300 | 0.300 | 0.294 | 0.292 | 0.292 |
| GT | Pomegranate | Zebra | Pomegranate | Pomegranate | Hummingbird |
| LDR | **Pomegranate** | **Zebra** | **Pomegranate** | **Pomegranate** | **Hummingbird** |
| PDS | 0.216 | 0.216 | 0.216 | 0.216 | 0.216 |

Figure 3: **Examples leading to the greatest and least disagreement**. We show the 5 most divergent and 5 least divergent samples according to the LDR ensemble. We measure prediction diversity with the Prediction Diversity Score (PDS) in §3.4. GT refers to the ground truth category. Ensemble predictions are shown in bold; in cases where ensemble members predict classes different from the ensemble prediction we provide them on the next line with standard font.

# 4 EXPERIMENTS

We present experiments that first evaluate the intrinsic diversification from LDR (§4.2) then evaluate several use cases of diverse ensembles for OOD detection (§4.3).

## 4.1 EXPERIMENTAL SETUP

**Baselines.** As a simple ensemble, we use a variant of *deep ensembles* (Lakshminarayanan et al., 2017), which uses models trained independently with different random seeds. To match the resource usage of our LDR, we also train only the last 2 layers of the models (i.e. they are "shallow ensembles"). We also consider simple ensembles of models with diverse hyperparameters (Wenzel et al., 2020). We reimplemented A2D (Pagliardini et al., 2023) and DivDis (Lee et al., 2023), with which we use unlabeled samples from ImageNet-R as disagreement data (the choice of dataset used for disagreement has little influence on the results, as seen in Table 9). For A2D, we use a frozen feature extractor and parallel training, i.e. all models are trained simultaneously rather than sequentially. See the implementation details in § A.5.

**Evaluation of OOD detection.** The task is to differentiate samples from the above OOD datasets against those from the ImageNet validation data (considered as ID). The evaluation includes both "semantic" and "covariate" types of shifts (Zhang et al., 2023; Hendrycks & Dietterich, 2019; Hendrycks et al., 2021a; Recht et al., 2019; Yang et al., 2024). Openimages-O (*OI* (Wang et al., 2022), 17k images, unlabeled) and iNaturalist (*iNat* (Huang & Li, 2021), 10k images, unlabeled) represent semantic shifts because their label sets are disjoint from ImageNet's. And ImageNet-C (*IN-C-i* or just *C-i* for corruption strength $i$ (Hendrycks & Dietterich, 2019), 50k images, 1k classes) represents a covariate shift because its label set is the same as ImageNet's but the style of images differs. We measure the OOD detection performance with the area under the ROC curve, following (Hendrycks & Gimpel, 2017).

## 4.2 DIVERSIFICATION

We start with the question of whether LDR truly diversifies the ensemble. To measure the diversity of the ensemble, we compute the number of unique predictions for each sample for the committee of models (#unique).

Table 1 shows the #unique and PDS values for the IN-Val as well as multiple OOD datasets. We observe that the deep ensemble baseline does not increase the diversity dramatically (e.g. 1.09 for IN-C-1) beyond no-diversity values (1.0). Diversification tricks like hyperparameter diversification (1.11 for IN-C-1) or A2D (1.04 for IN-C-1) only marginally change the prediction diversity. On the other hand, our LDR increases the prediction diversity across the board (e.g. 4.68 for iNat).

| Method | IN-Val | IN-C-1 | IN-C-5 | iNat | OI |
|---|---|---|---|---|---|
| DE | 1.05 | 1.09 | 1.19 | 1.31 | 1.23 |
| +Div. HPs | 1.04 | 1.11 | 1.32 | 1.48 | 1.33 |
| A2D | 1.11 | 1.04 | 1.15 | 1.19 | 1.91 |
| LDR (Ours) | **1.36** | **1.82** | **3.53** | **4.68** | **4.11** |

Table 1: **How diverse are ensembles?** We report the average #unique (number of unique classes among predictions of ensemble members for a given sample) on OOD datasets and IN-Val dataset (See §4.1 for the datasets). The ensemble size $M$ is 5 for all methods; $M$ is also the max possible #unique value.

Qualitative results on ImageNet-R further verify the ability of LDR to diversify the ensemble (Figure 3). As a measure for diversity, we use the Predictive Diversity Score (PDS) in §3.4. We observe that the samples inducing the highest diversity (high PDS scores) are indeed ambiguous: for the first image, where the "cowboy hat" is the ground truth category, we observe that "comic book" is also a valid label for the image style. On the other hand, samples with low PDS exhibit clearer image-to-category relationship.

## 4.3 OOD DETECTION

We study the impact of ensemble diversification on OOD detection capabilities of an ensemble as motivated in § 2. Once an ensemble is trained, we compute the epistemic uncertainty, or likelihood of the sample being OOD, following two schemes, $\eta_{\text{BMA}}$ and $\eta_{\text{PDS}}$ introduced in §3.4.

**LDR and PDS together lead to superior OOD detection performance.** We show the OOD detection results in Table 2. We mainly compare PDS to BMA because the latter is considered the most competitive baseline (Mukhoti et al., 2023) for uncertainty quantification. Comparison to other popular OOD detection baselines (Liu et al., 2020; Xia & Bouganis, 2022) can be seen in the Table 6. Results for ResNet18 (He et al., 2016) backbone can be seen in the Table 5. For the BMA scores, deep ensemble remains a strong baseline. In particular, when the hyperparameters are varied ("+Diverse HPs"), the

| Method | Unc. score | C-1 | C-5 | iNat | OI |
|---|---|---|---|---|---|
| 1 model | BMA | 61.5 | 83.3 | 95.8 | 90.9 |
| DE | BMA | 61.9 | 83.5 | 95.8 | 91.1 |
| +Div. HPs | BMA | **64.2** | **86.1** | **96.9** | **92.3** |
| DivDis | BMA | 59.8 | 84.3 | 96.6 | 92.2 |
| A2D | BMA | 59.4 | 83.5 | 96.6 | 91.6 |
| LDR | BMA | 64.1 | 84.5 | 96.0 | 91.5 |
| DE | PDS | 56.5 | 62.5 | 59.2 | 58.9 |
| +Div. HPs | PDS | 64.3 | 84.9 | 92.6 | 88.9 |
| DivDis | PDS | 60.0 | 85.1 | 96.9 | 93.9 |
| A2D | PDS | 59.9 | 85.0 | 97.1 | 93.9 |
| LDR | PDS | **68.1** | **89.4** | **97.7** | **94.1** |

Table 2: **OOD detection via ensembles.** For each OOD dataset (IN-C-1, IN-C-5, iNaturalist, and OpenImages), the ensembles are tasked to detect the respective OOD samples among IN-Val samples (ImageNet validation split). We show the AUROC scores for the OOD detection task. Ensemble size is fixed at $M = 5$. "Uncertainty score" refers to the epistemic uncertainty computation framework discussed in §3.4. See Appendix A.8 for comparison with other uncertainty scores.

detection AUROC reaches the maximal performance among the ensembles using the BMA scores. The quality of PDS is more sensitive to the ensemble diversity, as seen in the jump from the deep ensemble (e.g. $58.9\%$ for OpenImages) to the diverse-HP variant ($88.9\%$). However, when the ensemble is sufficiently diverse, such as when trained with LDR, the PDS leads to high-quality OODness scores. LDR with PDS achieves the best AUROC across the board, including the BMA variants.

**More OOD detection baselines.** To compare our method to the extended list of OOD detection baselines we follow the setup from (Mucsányi et al., 2024) for ImageNet C with severity level 5.

The details on the baselines can be seen in (Mucsányi et al., 2024). All of them use ResNet50 (He et al., 2016) as a backbone. For each method where it is applicable, we report the

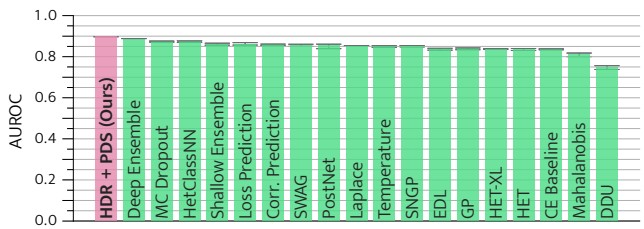

Figure 4: **OOD detection with extended baselines.** AUROC for OOD detection on ImageNet-C with severity level 5.

best result among both BMA and PDS uncertainty scores except for LDR where we only use PDS. Adopting the latest benchmark from Mucsányi et al. (2024) in Figure 4, we also conclude that our LDR+PDS achieves the state-of-the-art results.

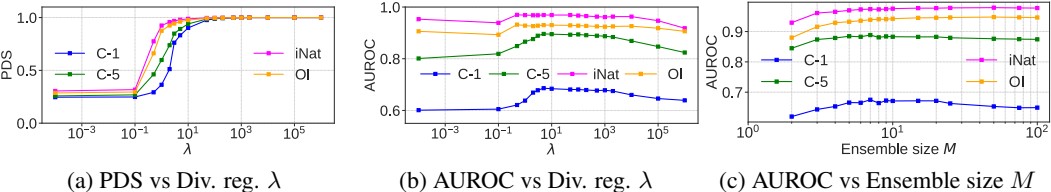

(a) PDS vs Div. reg. $\lambda$      (b) AUROC vs Div. reg. $\lambda$      (c) AUROC vs Ensemble size $M$

Figure 5: **Factor analysis for OOD detection**. We show the model answer diversity, measured by PDS, and the OOD detection performance, measured by AUROC, against $\lambda$ values, the loss weight for the disagreement regularizer, and the ensemble size $M$.

**Influence of diversification strength ($\lambda$).** We further study the impact of ensemble diversification on the OOD detection with the PDS estimator. In Figure 5, we observe that strengthening the diversification objective (higher $\lambda$) indeed leads to greater diversity (higher PDS), with a jump at around $\lambda \in [10^{-1}, 10^1]$. This range corresponds to the jump in the OOD detection performance (higher AUROC).

**Influence of ensemble size.** How ensemble size influences performance of our method? We can see that increasing ensemble size helps to improve AUROC for OOD detection on IN-C-1 (Figure 5). On other datasets increasing ensemble size only marginally helps, but using 5 models provides already a significant improvement over the smallest possible ensemble of size 2. It is also important to mention, that LDR framework is computationally more efficient w.r.t. ensemble size $M$ than for the previous methods such as A2D and DivDis: since we train ensembles for the fixed number of epochs, training complexity for LDR is $O(1)$ thanks to stochastic model pairs selection, while for A2D and DivDis it is $O(M^2)$.

**Limitations.** LDR cannot be applied to single-model setting. We do not consider controlled small-scale or adversarial settings where the training data lacks OOD samples. The method's sensitivity to label noise and model calibration is not examined and lies beyond the scope of this paper. Our study is limited to the vision domain; extending it to other modalities remains future work.

## 5 CONCLUSIONS

Treating training data as identical to ID is misleading at scale, where ID and OOD overlap. Ensemble diversification helps separate them via member disagreement, but prior methods were costly and required OOD data. We propose the Loss-guided Diversification Regularizer (LDR), which selects high-loss training samples to induce disagreement, removing the need for OOD data and reducing complexity from quadratic to constant. At ImageNet scale, LDR achieves state-of-the-art OOD detection, amplified by our Predictive Diversity Score (PDS).

## DISCLAIMER FOR USE OF LLMS

LLMs supported coding (experimentation, plotting) and writing refinement. The final version was carefully reviewed and finalized by the authors. LLMs were not used for ideation or experimental design.

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

# A    APPENDICES

## A.1    EXTENDED RELATED WORK ON ENSEMBLE DIVERSIFICATION

We review prior work on ensemble diversification from two different angles: (i) regularizer-based diversification, and (ii) diversification without altering the training objective.

**Application of ensemble disagreement to OOD detection**    Several studies have shown that ensemble disagreement provides a useful signal for OOD detection (Pagliardini et al., 2023; Lee et al., 2023; Tifrea et al., 2022; Lakshminarayanan et al., 2017; Everett et al., 2022; Linmans et al., 2020), since disagreement between ensemble components reflects epistemic uncertainty that tends to be higher on OOD samples than on ID samples (Mukhoti et al., 2023). This goes in line with the results on uncertainty estimation benchmarks that also highlight the efficacy of disagreement-based scores for OOD detection (Mucsányi et al., 2024; de Jong et al., 2024). Yet, no prior work has demonstrated how to train ensembles on ImageNet-scale data such that their disagreement aligns with the theoretical definition of OOD. Existing methods either (i) do not provide any guarantees of disagreement being a good proxy for OOD (Tifrea et al., 2022; Lakshminarayanan et al., 2017; Everett et al., 2022; Linmans et al., 2020), or (ii) require an auxiliary OOD dataset to induce disagreement (Pagliardini et al., 2023; Lee et al., 2023). As a consequence, at an ImageNet-scale, the first are less effective (Xia & Bouganis, 2022) while the second are impossible to train, since boundaries between ID and OOD a blurred at that scale.

**Diversification through regularizers.**    In pursuit of model disagreement, various auxiliary training objectives have been proposed to encourage diversity across models' weights (D'Angelo & Fortuin, 2021; de Mathelin et al., 2023a;b; Wang & Ji, 2023), features (Chen et al., 2024; Yashima et al., 2022; Yong et al., 2024), input gradients (Ross et al., 2020; Teney et al., 2022a;b; Trinh et al., 2024), or outputs (D'Angelo & Fortuin, 2021; Lee et al., 2023; Liu & Yao, 1999; Pagliardini et al., 2023; Scimeca et al., 2023; Everett et al., 2022). D'Angelo & Fortuin (2021) showed that a regularizer that repulses the ensemble members' weights or outputs leads to ensembles with a better approximation of Bayesian model averaging. This idea was extended by repulsing features (Yashima et al., 2022) and input gradients (Trinh et al., 2024). Since ensemble are most useful when the errors of its members are uncorrelated (Krogh & Vedelsby, 1994), the closest of the above objective is to diversify their outputs. This cannot be guaranteed with other objectives such as weight diversity for example, since two models could implement the exact same function with different weights due to the many symmetries in the parameter space of neural networks. For this reason, this paper focuses on methods for output-space diversification (Lee et al., 2023; Pagliardini et al., 2023). These were also highlighted as state-of-the-art in a recent survey on diversification (Benoit et al., 2024).

**Diversification without modifying the training objective.**    Once the loss modification is not an option, the most straightforward way to obtain diverse models is to independently train them with different seeds (Deep Ensembles (Lakshminarayanan et al., 2017) and Bayesian extensions (Wilson & Izmailov, 2020)), hyperparameters (Wenzel et al., 2020), augmentations (Li et al., 2023), or architectures (Zaidi et al., 2021). A computationally cheaper approach is to use models saved at different points during the training (Huang et al., 2017) or models derived from the base model by applying dropout (Gal & Ghahramani, 2016) or masking (Durasov et al., 2021). The "mixture of experts" paradigm (Zhou et al., 2018) can also be viewed as an ensemble where diversification happens by assigning different training samples to different ensemble members. Our experiments use Deep Ensembles (Lakshminarayanan et al., 2017) and ensembles of models trained with different hyperparameters (Wenzel et al., 2020) as baselines since they are strong approaches to OOD detection (Ovadia et al., 2019) and OOD generalization especially when combined with "model soups" (Wortsman et al., 2022).

## A.2    OOD GENERALIZATION

In this section, we present experiments that first evaluate use cases of diverse ensembles for OOD generalization.

| Method | $M$ | Arch. | $\mathcal{D}_{\text{div}}$ | Prediction ensemble | | | | | Uniform Soup | | | | | Oracle Selection | | | | |
|---|---|---|---|---|---|---|---|---|---|---|---|---|---|---|---|---|---|---|
| | | | | Val | A | R | C-1 | C-5 | Val | A | R | C-1 | C-5 | Val | A | R | C-1 | C-5 |
| 1 model | 1 | Deit3b | - | 85.4 | 37.9 | 44.7 | 75.6 | 38.5 | 85.4 | 37.9 | 44.7 | 75.6 | 38.5 | 85.4 | 37.9 | 44.7 | 75.6 | 38.5 |
| DE | 5 | Deit3b | - | **85.4** | 39.9 | 46.3 | 75.7 | 38.6 | **85.3** | 36.7 | 44.6 | 75.5 | 38.3 | **85.4** | 37.9 | 44.9 | 75.7 | 38.6 |
| +Div. HPs | 5 | Deit3b | - | **85.4** | 39.9 | 46.5 | 76.0 | 39.0 | **85.3** | 35.3 | 44.1 | 75.9 | 38.7 | **85.4** | 38.5 | 45.4 | 77.4 | 40.7 |
| DivDis | 5 | Deit3b | IN-R | 85.1 | 36.3 | 41.8 | 77.2 | 40.2 | 84.8 | **40.7** | 42.5 | 76.2 | 38.9 | 85.2 | 35.8 | 40.8 | 77.2 | 40.2 |
| A2D | 5 | Deit3b | IN-R | 85.1 | 37.8 | 45.2 | 77.2 | 40.3 | 84.5 | 39.3 | 45.1 | 75.5 | 39.1 | 85.2 | 36.6 | 44.3 | 77.3 | 40.4 |
| LDR | 5 | Deit3b | $\alpha_n \uparrow$ | 85.3 | **43.0** | **48.7** | 77.3 | 40.7 | **85.3** | 40.3 | **46.1** | 77.3 | 40.6 | 85.1 | 38.3 | 45.3 | 77.2 | 40.4 |
| DE | 50 | Deit3b | - | **85.5** | 38.8 | 45.8 | 75.6 | 38.5 | **85.4** | 37.5 | 45.0 | 75.5 | 38.4 | **85.5** | 38.1 | 45.2 | 75.7 | 38.6 |
| +Div. HPs | 50 | Deit3b | - | 85.5 | 42.5 | 48.5 | **76.0** | 39.0 | **85.4** | 36.4 | 44.8 | **75.9** | 38.8 | 85.5 | 38.5 | 45.6 | **77.5** | 40.8 |
| LDR | 50 | Deit3b | $\alpha_n \uparrow$ | 83.6 | **50.6** | **53.8** | 75.8 | **39.3** | 83.5 | **39.2** | **46.5** | 75.8 | **39.3** | 82.6 | **39.0** | **45.8** | 74.4 | 38.3 |
| DE | 5 | RN18 | - | **69.8** | 0.5 | **20.8** | 51.9 | 14.6 | **69.8** | 0.4 | 19.4 | **51.9** | 14.6 | **69.8** | 0.4 | 19.5 | **51.9** | 14.6 |
| LDR | 5 | RN18 | $\alpha_n \uparrow$ | 69.6 | 0.6 | **20.8** | 51.8 | 14.6 | 69.6 | 0.5 | **19.6** | 51.8 | **14.6** | 69.7 | 0.5 | **19.6** | **51.9** | **14.6** |

Table 3: **OOD generalization of ensembles.** Models are trained on the ImageNet training split. $M$ is the ensemble size. $\mathcal{D}_{\text{div}}$ corresponds to samples on which the respective diversification objectives are applied. $\alpha_n \uparrow$ denotes samples with high $\alpha_n$ values (see § 3.3).

**Evaluation of OOD generalization.** We evaluate the classification accuracy of the ensembles trained on ImageNet with the (ID) validation split of ImageNet (IN-Val, 50,000 samples) and multiple OOD datasets: ImageNet-A (*A* (Hendrycks et al., 2021b), 7.5k images & 200 classes), ImageNet-R (*R* (Hendrycks et al., 2021a), 30k images, 200 classes), ImageNet-C.

Ensembles aggregate the outputs of multiple models to make more accurate predictions (Breiman, 1996; 2001; Hansen & Salamon, 1990; Ho, 1995; Krogh & Vedelsby, 1994) see § 2 for details. We hypothesize that the superior diversification ability verified in §4.2 leads to greater OOD generalization due to the consideration of more robust hypotheses that do not rely on obvious spurious correlations.

**Ensemble aggregation for OOD generalization.** As a means to exploit such robust hypotheses, we consider 3 aggregation strategies. (1) *Oracle selection*: the best-performing individual model is chosen from an ensemble (Pagliardini et al., 2023; Teney et al., 2022a). The final prediction is given by $f(x; \theta^{m^\star})$ where $m^\star := \arg\max_m \text{Acc}(f^m, \mathcal{D}^{\text{ood}})$. (2) *Prediction ensemble* is a vanilla prediction ensemble where the logit values are averaged: $\frac{1}{M} \sum_m f^m(x)$ (Wortsman et al., 2022). (3) *Uniform soup* (Wortsman et al., 2022) averages the weights themselves. The final prediction is given by $f(x; \frac{1}{M} \sum_m \theta^m)$.

**Stochastic sum improves OOD generalization for ensembles.** We show the OOD generalization performance of ensembles in Table 3, for the three ensemble prediction aggregation strategies described above. We observe that our framework (LDR) is superior in OOD generalization performance for the prediction ensemble and uniform soup while being on par with best baselines for the oracle selection. LDR is particularly strong in the prediction ensemble (e.g. 48.7% for $M = 5$ and 53.8% for $M = 50$ on ImageNet-R) and uniform soup (e.g. 46.1% for $M = 5$ and 46.5% for $M = 50$ on ImageNet-R). We contend that the increased ensemble diversity contributes to the improvements in OOD generalization, however, it is worth noting the the best results are achieved by diversifying ensembles solely with stochastic sum while keeping $\lambda = 0$. We also remark that the LDR framework (LDR) envelops the performance of A2D and DivDis in this experiment. Together with the superiority of computational efficiency (as discussed at the end of § 4.3) of LDR over the previous diversification methods, this demonstrates that LDR provides a scalable solution for ensemble diversification on ImageNet scale.

**Deep ensembles are a strong baseline.** We also note that deep ensemble, particularly with diverse hyperparameters, provides a strong baseline, outperforming dedicated diversification methodologies under the oracle selection strategy when $M = 5$. It also provides a good balance between ID (ImageNet validation split) and OOD generalization.

## A.3 DISAGREEMENT-BASED CHARACTERIZATION OF OOD

This section shows that, for ensembles making no ID errors, disagreement on a sample implies the sample is OOD. In the following subsections we outline the setup (§ A.3.1), state key assumptions (§ A.3.2), and conclude with the main theorem (§ A.3.3).

### A.3.1 SETUP.

Let $\mathcal{X}$ be the input space and $\mathcal{Y}$ a finite label set. Let $P$ be a distribution on $\mathcal{X} \times \mathcal{Y}$, $\varepsilon > 0$ be some small threshold and define the support as $\mathrm{supp}_\varepsilon(P) := \{(x, y) \in \mathcal{X} \times \mathcal{Y} : P(x, y) > \varepsilon\}$ (we fix one $\varepsilon$ throughought the paper and call $\mathrm{supp}_\varepsilon$ as $\mathrm{supp}$ for brevity). Let $\mathcal{F} \subseteq \mathcal{Y}^{\mathcal{X}}$ be a hypothesis class (e.g., a VC class).

**Definition 2** (Solutions). *The set of $\underline{\text{solutions}}$ (zero $P$-risk classifiers) is*

$$\mathcal{S} := \Big\{ f \in \mathcal{F} : R_P := \Pr_{(x,y) \sim P} \big[ f(x) \neq y \big] = 0 \Big\}.$$

**Definition 3** (Disagreement indicator). *Given a set $\mathcal{A} \subseteq \mathcal{F}$, define*

$$J_{\mathcal{A}}(x) := \mathbf{1}\{\exists f, g \in \mathcal{A} \text{ with } f(x) \neq g(x)\}.$$

*We will use $J_{\mathcal{S}}$ (all solutions) and, for an ensemble $\{f_1, \ldots, f_M\} \subseteq \mathcal{S}$, the finite version $J_M(x) := \mathbf{1}\{\exists m \neq m' : f^m(x) \neq f^{m'}(x)\}$.*

### A.3.2 ASSUMPTIONS.

**Assumption 1** (Realizability). *There exist at least two zero-risk classifiers: $|\mathcal{S}| > 1$.*

**Assumption 2** (Off-support extension richness). *For every $(x, y) \notin \mathrm{supp}(P)$, there exist $f, g \in \mathcal{S}$ such that $f(x) \neq g(x)$.*

Remark: Assumption 2 holds for many standard VC classes that can locally alter predictions outside a compact set containing $\{(x, y) \in \mathrm{supp}(P)\}$ while preserving $R_P = 0$ (e.g., linear separators when $x$ lies outside the region containing $\mathrm{supp}(P)$, decision trees, or sufficiently expressive neural networks).

### A.3.3 THEOREM

We show that, for ensembles with zero ID errors, disagreement precisely characterizes OOD inputs.

**Theorem 3** (Equivalence of distributional and disagreement-based definitions of OOD). *Under Assumptions 1–2:*

$$\begin{array}{lll}
\textit{(ID)} & (x, y) \in \mathrm{supp}(P) \iff & J_M(x) = 0, \\
\textit{(OOD)} & (x, y) \in (\mathcal{X} \times \mathcal{Y}) \setminus \mathrm{supp}(P) \iff & J_M(x) = 1.
\end{array}$$

*Thus, disagreement exactly characterizes out-of-distribution inputs at the solution level.*

*Proof.* **Case 1 (In-distribution).** Suppose $(x, y) \in \mathrm{supp}(P)$. If two solutions $f, g \in \mathcal{S}$ disagreed at $x$, then at least one of $f$ or $g$ would differ from $y$, yielding nonzero risk. This contradicts $f, g \in \mathcal{S}$. Hence $J_M(x) = 0$.

**Case 2 (Out-of-distribution).** Suppose $(x, y) \notin \mathrm{supp}(P)$ which is equivalent to $(x, y) \in (\mathcal{X} \times \mathcal{Y}) \setminus \mathrm{supp}(P)$. By Assumption 2, there exist two solutions $f, g \in \mathcal{S}$ that differ on $x$. Hence $J_M(x) = 1$.

The bicondition follows by the same contradictions in the reverse direction. $\square$

## A.4 DISAGREEMENT ON HIGH LOSS SAMPLES INCREASES MODEL DISAGREEMENT ON OOD SAMPLES

This section proves that inducing ensemble disagreement on high-loss samples from the training dataset increases disagreement on OOD samples. Since models agree on ID, this enables using disagreement as an OOD uncertainty score. In the following subsections we outline the setup with high-loss samples, define pairwise disagreement (§ A.4.1), state key assumptions (§ A.4.2), prove supporting lemmas (§ A.4.3), and conclude with the main theorem (§ A.4.4).

### A.4.1 SETUP

Let $\mathcal{X}$ be the input space and $\mathcal{Y}$ a finite label set. We consider ID and OOD distributions $P$ and $Q$ with a training set sampled from a mixture of them: $(1 - \varepsilon)P + \varepsilon Q$ (small $\varepsilon$). Let $\mathcal{F} \subseteq \mathcal{Y}^{\mathcal{X}}$ be the model family, and let $\mathcal{E} := \{f^1, \ldots, f^M\} \subset \mathcal{F}$ be a finite ensemble. We use the ensemble cross-entropy loss $\sum_{m=1}^{M} -\log p_y^m(x)$. Let $H \subset \mathcal{D}$ denote high-loss points, i.e., there exists $\gamma > 0$ such that $\sum_{m=1}^{M} -\log p_y^m(x) \geq \gamma$ for all $(x, y) \in H$, and write $\mu := \mathrm{Unif}(H)$. Define pairwise disagreement $I_{m,m'}(x) := \mathbf{1}\{f^m(x) \neq f^{m'}(x)\}$ and existential disagreement $J(x) := \mathbf{1}\{\exists m \neq m' : f^m(x) \neq f^{m'}(x)\}$. We will use metric balls $B_r(x) := \{x' \in \mathcal{X} : d(x', x) < r\}$ for radius $r > 0$ under a metric $d$ on $\mathcal{X}$. The disagreement on $H$ is $\mathrm{Dis}(\mathcal{E}; H) := \mathbb{E}_{x \sim \mu} \mathbb{E}_{m \neq m'}[I_{m,m'}(x)]$ with empirical counterpart $\widehat{\mathrm{Dis}}(\mathcal{E}; H) := \frac{1}{|H|} \sum_{x \in H} \frac{1}{M(M-1)} \sum_{m \neq m'} I_{m,m'}(x)$. Training enforces $\widehat{\mathrm{Dis}}(\mathcal{E}; H) \geq \delta_H$. Our goal is to bound the OOD disagreement probability $\mathrm{Pr}_{x \sim Q}[J(x) = 1] = \mathrm{Pr}_{x \sim Q}[\exists m \neq m' : f^m(x) \neq f^{m'}(x)]$.

### A.4.2 ASSUMPTIONS

**Assumption 3** (Clustered OOD near observed OOD.). *There exists a radius $r > 0$ and constants $\alpha_0 > 0$ such that:*

*For any $x \in \mathrm{supp}(Q)$,*
$$Q(B_r(x)) \geq \alpha_0.$$

*Informally: we occasionally sample OOD points in training, and each such point lies in an OOD cluster with nontrivial Q-mass; high loss points are enriched with OOD.*

**Assumption 4** (Among points of disagreement, a fraction of at least $\rho$ is truly from $Q$). *There exists $\rho \in (0, 1]$ such that*
$$\mathrm{Pr}[X \in Q \mid J(X) = 1] \geq \rho.$$

**Assumption 5** (Local label stability for the ensemble). *There exists $\bar{\eta} \in [0, 1)$ such that, for $\mu$-a.e. $x$ and for a random independent draw $x' \sim Q$,*
$$\mathrm{Pr}\left[J(x') = 0 \,\middle|\, x' \in B_r(x),\, J(x) = 1\right] \leq \eta(x),$$

*with*
$$\bar{\eta} := \mathbb{E}_{x \sim \mu}[\eta(x)].$$

*(If the ensemble disagrees at $x$, then with high probability it still disagrees in the $r$-ball around $x$. This can be justified e.g. by margin + Lipschitz conditions on logits for the disagreeing models.)*

**Assumption 6** (Generalization of disagreement from empirical to $\mu$).). *With probability $\geq 1 - \delta$ over the training sample, there exists a complexity term $\Gamma \geq 0$ (Rademacher/PAC-Bayes-like) such that*
$$\left|\widehat{\mathrm{Dis}}(\mathcal{F}; H) - \mathrm{Dis}(\mathcal{F}; H)\right| \leq \Gamma.$$

*Equivalently,*

$$\mathrm{Dis}(\mathcal{F}; H) = \mathbb{E}_{x \sim \mu} \mathbb{E}_{m \neq m'}[I_{m,m'}(x)] \geq \delta_H - \Gamma. \tag{11}$$

*because training enforces $\widehat{\mathrm{Dis}}(\mathcal{F}; H) \geq \delta_H$ as stated in § A.4.1.*

### A.4.3 LEMMAS

**Lemma 1.** *High loss is a necessary condition for disagreement.*

$$J(x) = 1 \implies \exists \tau > 0 : \sum_{m}^{M} -\log p_y^m(x) \geq \tau.$$

*Proof.* $J(x) = 1$ means that at least one model $f^m$ disagrees with the majority prediction on $x$.

Then, at least one $f^m(x)$ puts most of its probability mass on some class $c \neq \hat{y}$. Therefore, the averaged prediction cannot be overly concentrated on the true label $y$; the ensemble cross-entropy $\sum_m^M - \log p_y^m(x)$ cannot be arbitrarily small.

Then, there exists a constant $\tau > 0$ such that

$$J(x) = 1 \implies \sum_m^M - \log p_y^m(x) \geq \tau.$$

Equivalently if we set $\gamma = \tau$ in the definition of $H$ in § A.4.1,

$$\Pr(x \notin H \mid J(x) = 1) = 0.$$

$\square$

This captures the intuition: disagreement requires at least one model to be confident in another class, so the ensemble cross-entropy cannot be very low. This lemma aligns well with Proposition 5.2 in (Mukhoti et al., 2023).

Since disagreement points lie inside $H$ (up to $\tau$), the bound in Assumption 4 holds when conditioning on $H$:

**Corollary 1.** *If $\gamma = \tau$*

$$\Pr[X \in Q \mid J(X) = 1, X \in H] \geq \rho, \rho > 0.$$

**Lemma 2** (Pairwise vs. existential disagreement). *For every $x$,*

$$\mathbb{E}_{m \neq m'}[I_{m,m'}(x)] \leq J(x).$$

*Proof.* If no pair disagrees, then $I_{m,m'}(x) = 0$ for all $m \neq m'$, so the LHS is 0 and equals $J(x) = 0$. If some pair disagrees, then $J(x) = 1$ while each $I_{m,m'}(x) \in \{0, 1\}$, so their average is $\leq 1 = J(x)$.

Averaging over $x \sim \mu$,

$$\mathbb{E}_{x \sim \mu}[J(x)] \geq \mathbb{E}_{x \sim \mu} \mathbb{E}_{m \neq m'}[I_{m,m'}(x)] \geq \delta_H - \Gamma \quad \text{(by Assumption 6).}$$

$\square$

**Lemma 3** (A coupling lower bound for OOD coverage). *Let*

$$\mathcal{U} := \bigcup_{x:J(x)=1} B_r(x).$$

*Then*

$$Q(\mathcal{U}) \geq \mathbb{E}_{x \sim \mu}[J(x) Q(B_r(x))].$$

*Proof.* Consider independent $x \sim \mu$ and $x' \sim Q$. The event

$$\{ x' \in B_r(x) \wedge J(x) = 1 \}$$

implies

$$\{ x' \in \mathcal{U} \}.$$

Hence

$$\Pr[x' \in \mathcal{U}] \geq \Pr[x' \in B_r(x) \wedge J(x) = 1] = \mathbb{E}_{x \sim \mu}[J(x) Q(B_r(x))].$$

But $\Pr[x' \in \mathcal{U}] = Q(\mathcal{U})$, which completes the proof. $\square$

**Lemma 4** (Turn the expectation into a constant). *Under Assumptions 3, 4,*

$$Q(\mathcal{U}) \geq \mathbb{E}_{x \sim \mu}[J(x) Q(B_r(x))] \geq \alpha_0 \rho \, \mathbb{E}_{x \sim \mu}[J(x)]. \tag{12}$$

*Proof.* Decompose by whether $x \in Q$ or not:

$$\mathbb{E}_{x \sim \mu}[J(x) \, Q(B_r(x))] = \mathbb{E}\big[J(x) \, Q(B_r(x)) \, \mathbf{1}_{\{x \in Q\}}\big] + \mathbb{E}\big[J(x) \, Q(B_r(x)) \, \mathbf{1}_{\{x \notin Q\}}\big].$$

Drop the nonnegative second term and use $Q(B_r(x)) \geq \alpha_0$ for $x \in Q$ (Assumption 3):

$$\geq \alpha_0 \, \mathbb{E}\big[J(x) \, \mathbf{1}_{\{x \in Q\}}\big] = \alpha_0 \, \Pr[x \in Q \wedge J(x) = 1].$$

Now,

$$\Pr[x \in Q \wedge J(x) = 1] = \Pr[x \in Q \mid J(x) = 1] \cdot \Pr[J(x) = 1] \, \geq \, \rho \, \mathbb{E}_{x \sim \mu}[J(x)] \quad \text{by Corollary 1}$$

Combining the inequalities, we obtain

$$Q(\mathcal{U}) \, \geq \, \alpha_0 \rho \, \mathbb{E}_{x \sim \mu}[J(x)].$$

$\square$

**Lemma 5** (Witness-based lower bound under local stability). *Let $\mu = \mathrm{Unif}(H)$ be the uniform distribution over high loss points, let $J(x) = \mathbf{1}\{\exists m \neq m' : f^m(x) \neq f^{m'}(x)\}$ denote the disagreement indicator, and for radius $r > 0$ define*

$$\mathcal{U} := \bigcup_{x \in H : J(x) = 1} B_r(x).$$

*Define the "failure" and "success" subsets of $\mathcal{U}$ by*

$$\mathcal{G} := \{x' \in \mathcal{U} : J(x') = 0\}, \qquad \mathcal{U}_{\mathrm{dis}} := \{x' \in \mathcal{U} : J(x') = 1\}.$$

*Then, under Assumption 5 (local stability),*

$$Q(\mathcal{U}_{\mathrm{dis}}) \, \geq \, (1 - \bar{\eta}) \, Q(\mathcal{U}) \geq Q(\mathcal{U}) - \bar{\eta},$$

*where $\bar{\eta} := \mathbb{E}_{x \sim \nu}[\eta(x)]$ and $\nu$ is the witness distribution defined below.*

*Proof.* For each $x' \in \mathcal{U}$, choose a witness center $w(x') \in H$ such that $J(w(x')) = 1$ and $x' \in B_r(w(x'))$. (If multiple centers cover $x'$, pick one by a fixed rule; if $x' \notin \mathcal{U}$, leave $w(x')$ undefined.) This induces a distribution $\nu$ on centers by pushing forward $Q(\cdot \mid x' \in \mathcal{U})$ through $w$:

$$\nu(A) := \Pr_{x' \sim Q}\big[w(x') \in A \mid x' \in \mathcal{U}\big].$$

By conditioning on witnesses,

$$\begin{aligned} Q(\mathcal{G}) &= Q(\mathcal{U}) \Pr[J(x') = 0 \mid x' \in \mathcal{U}] \\ &= Q(\mathcal{U}) \, \mathbb{E}_{x \sim \nu} \Pr_{x' \sim Q}[J(x') = 0 \mid x' \in B_r(x)]. \end{aligned}$$

For each witness $x$ with $J(x) = 1$,

$$\Pr_{x' \sim Q}[J(x') = 0 \mid x' \in B_r(x)] \, \leq \, \eta(x).$$

Hence,

$$Q(\mathcal{G}) \, \leq \, Q(\mathcal{U}) \, \mathbb{E}_{x \sim \nu}[\eta(x)] = Q(\mathcal{U}) \, \bar{\eta}.$$

Since $Q(\mathcal{U}) = Q(\mathcal{U}_{\mathrm{dis}}) + Q(\mathcal{G})$,

$$Q(\mathcal{U}_{\mathrm{dis}}) = Q(\mathcal{U}) - Q(\mathcal{G}) \, \geq \, (1 - \bar{\eta}) \, Q(\mathcal{U}).$$

Note that $Q(\mathcal{U}) \leq 1$, therefore

$$Q(\mathcal{G}) \leq \bar{\eta}, \quad \text{hence} \quad Q(\mathcal{U}_{\mathrm{dis}}) \geq Q(\mathcal{U}) - \bar{\eta}.$$

$\square$

### A.4.4 FINAL THEOREM

We now prove that inducing disagreement on high-loss samples from training dataset amplifies ensemble disagreement on OOD samples.

**Theorem 4** (Lower bound on OOD existential disagreement). *Under Assumptions 3-6, the ensemble satisfies*

$$\Pr_{x \sim Q}\Big[\exists m \neq m' : f^m(x) \neq f^{m'}(x)\Big] \geq \alpha_0 \rho (\delta_H - \Gamma) - \bar{\eta}.$$

*Proof.* From Lemma 5 and inequality 12,

$$Q(\mathcal{U}_{\text{dis}}) \geq \alpha_0 \rho \, \mathbb{E}_{x \sim \mu}[J(x)] - \bar{\eta}.$$

By Lemma 2 and inequality 11,

$$\mathbb{E}_{x \sim \mu}[J(x)] \geq \mathbb{E}_{x \sim \mu} \, \mathbb{E}_{m \neq m'}[I_{m,m'}(x)] \geq \delta_H - \Gamma.$$

Combining the two bounds gives

$$Q(\mathcal{U}_{\text{dis}}) \geq \alpha_0 \rho (\delta_H - \Gamma) - \bar{\eta}.$$

Finally, since $\mathcal{U}_{\text{dis}} \subseteq \{x : J(x) = 1\}$, we have

$$Q(\mathcal{U}_{\text{dis}}) \leq \Pr_{x \sim Q}[J(x) = 1] = \Pr_{x \sim Q}\Big[\exists m \neq m' : f^m(x) \neq f^{m'}(x)\Big].$$

Therefore,

$$\Pr_{x \sim Q}\Big[\exists m \neq m' : f^m(x) \neq f^{m'}(x)\Big] \geq \alpha_0 \rho (\delta_H - \Gamma) - \bar{\eta}.$$

$\square$

## A.5 IMPLEMENTATION DETAILS

For both OOD detection and OOD generalization tasks, we train an ensemble of models with LDR using the AdamW optimizer (Loshchilov & Hutter, 2019), a batch size varies from 16 to 256, learning rate from $10^{-4}$ to $10^{-3}$, weight decay is fixed to 0.01, and number of epochs to 10. The diversity weight $\lambda$ varies from 0 to 5 and the stochastic pairing is done for $|\mathcal{I}| = 2$ models for each mini-batch. All experiments use models based on the Deit3b architecture (Touvron et al., 2022) pretrained on ImageNet21k (Deng et al., 2009). As suggested in §A.5.1 we train only the last 2 layers. As in-domain (ID) data we use the training split of ImageNet ($|\mathcal{D}| = 1,281,167$). All experiments were run on RTX2080Ti GPUs with 12GB vRAM and 40GB RAM. Each experiment took between 2 to 12 hours.

### A.5.1 STOCHASTIC SUM AND SHALLOW DISAGREEMENT

To improve the scalability of our **Loss-guided Diversification Regularizer** (**LDR**), we introduce a relaxation of the exhaustive pairwise comparisons between all the models in the ensemble (see the second term of Equation 6). This has not been a problem in previous work, as the considered ensemble sizes were modest ($M = 2$ for Lee et al. (2023) and $M = 5$ for Pagliardini et al. (2023)). However, the computation scales in $O(M^2)$ and becomes a problem when the ensemble grows as large as $M = 50$, as in our experiments (Table 3).

**Stochastic sum.** To overcome this quadratic scaling law we propose to use a stochastic sum. For every mini-batch $B$, we use a random subset of models $|\mathcal{I}| \in \{1, \cdots, M\}$ on which to compute the diversification term in Equation 6 (please see Appendix A.13 for mathematical justification). In our experiments, we randomly sample one pair of models per batch ($|\mathcal{I}| = 2$). Such stochastic sum size allows to reduce training of an ensemble of 50 models from theoretical 663 GPU hours per epoch to 30 minutes per epoch (see Appendix A.12). In addition to the computational benefits, stochastic sums contribute to the ensemble diversity by exposing each member to different subsets of training data.

**Shallow disagreement.** To further speed up the training, we consider updating only a subset of the layers of the model with the LDR objective, keeping others frozen. More specifically, each ensemble member in the experiments of Section 4 is based on a frozen Deit3b model (Touvron et al., 2022) of which we diversify only the last two layers. Diversifying only the last layer results in worse performance presumably due to the convexity of the optimization problem (see Appendix A.6).

## A.6 VARYING THE NUMBER OF TRAINABLE LAYERS

To perform a sensitivity study to the number of layers diversified for each ensemble member we trained only one last layer of DeiT-3b and compared it to the ensemble from the main experiments with the last two layers trained. Both ensembles have size 5 and were trained on the ImageNet training split. The results can be seen in Table 4. Generalization performance did not change much, with the biggest change for ImageNet-C with the corruption strength 5 where ensemble accuracy dropped from $40.8\%$ for one layer to $40.6\%$ for two layers. However, OOD detection performance is better across the board for the case when two layers are diversified, for example, the detection AUROC scores for one layer diversified vs two layers diversified are $0.928$ vs $0.941$ for OpenImages and $0.964$ vs $0.977$ for iNaturalist. We believe that it can be explained by the fact that when one linear layer is trained with cross-entropy loss the optimization problem becomes convex making it harder for disagreement regularizer to promote diversity for different solutions, i.e. ensemble members tend to have similar weight matrices and disagree on OOD samples less.

| # Layers | Ensemble Acc. | | | | | AUROC | | | |
| --- | --- | --- | --- | --- | --- | --- | --- | --- | --- |
| | Val | IN-A | IN-R | C-1 | C-5 | C-1 | C-5 | iNat | OI |
| 1 | 85.2 | 42.3 | **48.2** | **77.3** | **40.8** | 0.677 | 0.889 | 0.964 | 0.928 |
| 2 | **85.3** | **42.4** | 48.1 | **77.3** | 40.6 | **0.681** | **0.894** | **0.977** | **0.941** |

Table 4: Varying the number of trainable layers.

## A.7 RESNET18 RESULTS

To check the applicability of our method to other architectures we trained an ensemble of 5 models with the whole model but last layer frozen using ResNet18 as a feature extractor. We compared LDR with A2D disagreement regularizer and stochastic sum size $|\mathcal{I}| = 2$ vs deep ensemble in Table 5. Both ensembles were trained on the ImageNet training split. LDR shows better OOD detection performance across the board, for example, the detection AUROC scores for one deep ensemble vs LDR are $80.1$ vs $81.2$ for OpenImages and $86.4$ vs $87.3$ for iNaturalist.

| Method | Unc. score | C-1 | C-5 | iNat | OI |
| --- | --- | --- | --- | --- | --- |
| DE | BMA | 66.4 | 86.1 | **86.4** | 80.1 |
| LDR | BMA | **67.8** | **87.8** | 86.2 | **80.2** |
| DE | PDS | 64.4 | 77.4 | 75.0 | 76.1 |
| LDR | PDS | **68.6** | **86.0** | **87.3** | **81.2** |

Table 5: OOD detection results for ResNet18 backbone.

## A.8 COMPARISON OF PDS TO OTHER UNCERTAINTY SCORES

In this section, we evaluate how sensitive LDR is to the choice of uncertainty scores in Table 6.

### A.8.1 EPISTEMIC UNCERTAINTY SCORES

Alongside the scores in §3.4, we introduce the strongest alternatives to PDS for measuring epistemic uncertainty below.

| Method | Unc. score | C-1 | C-5 | iNat | OI |
|--------|-----------|-----|-----|------|-----|
| \multicolumn Bayesian model averaging (BMA) | | | | | |
| 1 model | BMA | 61.5 | 83.3 | 95.8 | 90.9 |
| LDR | BMA | 64.1 | 84.5 | 96.0 | 91.5 |
| \multicolumn Epistemic uncertainty scores | | | | | |
| LDR | PDS | 68.1 | 89.4 | **97.7** | **94.1** |
| LDR | $\overline{E}(f)$ | 63.3 | 85.8 | **97.7** | 90.8 |
| LDR | $\overline{H}(p)$ | 58.0 | 82.5 | 96.0 | 91.6 |
| LDR | $\overline{p}$ | **67.3** | **87.4** | 80.9 | 82.9 |
| LDR | Ens. $H(p)$ | 58.0 | 82.6 | 96.0 | 91.6 |

Table 6: **OOD detection via ensembles.** For each OOD dataset (IN-C-1, IN-C-5, iNaturalist, and OpenImages), the ensembles are tasked to detect the respective OOD samples among IN-Val samples (ImageNet validation split). We show the AUROC scores for the OOD detection task. Ensemble size is fixed at $M = 5$. "Uncertainty score" refers to the epistemic uncertainty computation framework discussed in §3.4 as well as other methods for OOD detection discussed in Appendix A.8.

**Average Energy ($\overline{E}(f)$)**  We compute the energy uncertainty score (Liu et al., 2020) for each ensemble member and then average energy values among ensemble members (we omit the temperature term $T$ from the original definition by setting it always equal to 1):

$$\overline{E}(f) = -\frac{1}{M} \sum_{m=1}^{M} \log \sum_{c=1}^{C} e^{f_c^m(\mathbf{x})} \tag{13}$$

**Average Entropy ($\overline{H}(p)$) and Ensemble Entropy (Ens. $H(p)$):**

$$\overline{H}(p) = \frac{1}{M} \sum_{m=1}^{M} \mathcal{H}\left[p^m(x)\right] \tag{14}$$

$$\text{Ens. } H(p) = \mathcal{H}\left[\frac{1}{M} \sum_{m=1}^{M} p^m(x)\right], \tag{15}$$

where $\mathcal{H}[p(x)] = -\frac{1}{C} \sum_{c=1}^{C} p_c(x) \log p_c(x)$

**Average confidence of ensemble members ($\overline{p}$)**  :

$$\overline{p} = \frac{1}{M} \sum_{m=1}^{M} \max_c p_c^m(x) \tag{16}$$

### A.8.2   RESULTS

As can be seen in Table 6 PDS achieves the best OOD detection performance overall, notably on ImageNet-C-5 ($89.4$ vs. $87.4$) and OpenImages ($94.1$ vs. $91.6$).

### A.9    COMPARISON TO A TWO-STAGE APPROACH

To perform an ablation study on the way samples for disagreement are selected in Table 7 we compared an ensemble trained with Equation 6 (called "joint" in the table) against a 2-stage approach. Instead of disagreeing on all samples with adaptive weight $\alpha_n$ as in Equation 6 we first computed the confidence of the pre-trained DeiT-3B model on all samples in ImageNet training split and then selected samples with a confidence lower than $0.2$ which resulted in $18002$ samples (to approximately match the sizes of ImageNet-A and ImageNet-R). Then we trained an ensemble by minimizing A2D disagreement regularizer on these samples while minimizing cross-entropy on all other samples. Both ensembles had size 5 and stochastic sum size $|\mathcal{I}| = 2$. While such an approach might sound simpler, LDR is more straightforward and efficient, since there is no need to train an initial model to determine samples for disagreement. Both ensembles have a similar generalization performance, with the biggest difference for ImageNet-R where ensemble accuracy dropped from $48.5\%$ for 2-stage approach to $48.1\%$ for the joint. In contrast, OOD detection performance is significantly better across the board for the joint approach, for example, the detection AUROC scores are $0.845$ vs $0.896$ for ImageNet-C with corruption strength 5 and $0.911$ vs $0.941$ for OpenImages. We think that such a drastic difference in OOD detection performance can be caused by the fact that the set of samples selected for disagreement may be suboptimal which makes the confidence threshold (set as $0.2$ for this experiment) an important hyperparameter and adds even more complexity to the 2-stage approach.

| | | Ensemble Acc. | | | | AUROC | | | |
|---|---|---|---|---|---|---|---|---|---|
| Type | Val | IN-A | IN-R | C-1 | C-5 | C-1 | C-5 | iNat | OI |
| 2-stage | 85.2 | **42.4** | **48.5** | **77.3** | **40.7** | 0.597 | 0.845 | 0.960 | 0.911 |
| Joint | **85.3** | **42.4** | 48.1 | **77.3** | 40.6 | **0.686** | **0.896** | **0.977** | **0.941** |

Table 7: Comparison with a two-stage approach.

### A.10    SMALL-SCALE EXPERIMENTS

To check the performance of our method on the small-scale datasets, we conducted additional experiments on the Waterbirds dataset (Sagawa* et al., 2020) (Table 8), since both A2D and DivDis also provided results on it. We report the worst group (test) accuracy for ensembles of size 4. We trained A2D, DivDis, and an ensemble with LDR and A2D disagreement regularizer on Waterbirds training split. We did not use stochastic sum for LDR-A2D to factor out its influence. A2D and DivDis used the validation set for disagreement. While DivDis discovers a better single model having best accuracy of $87.2\%$ against $83.2\%$ for the proposed LDR-A2D method, the ensemble is clearly better with LDR-A2D: $80.6\%$ vs $78.3\%$ for DivDis.

| | Oracle selection | Ensemble |
|---|---|---|
| ERM | 76.5 | 72.0 |
| DivDis | **87.2** | 78.3 |
| A2D | 78.3 | 78.3 |
| LDR | 83.5 | **80.6** |

Table 8: Worst group test accuracy on Waterbirds

### A.11    OOD DATASETS FOR DISAGREEMENT

To analyze the influence of OOD data used for disagreement we performed additional experiments with ensemble members disagreeing on ImageNet-R and ImageNet-A in Table 9. We compare A2D and Div (Lee et al., 2023) diversification regularizers. Usage of ImageNet-A or ImageNet-R resulted in almost identical (identical after rounding) OOD generalization performance for A2D disagreement regularizer, while for Div regularizer ensemble accuracy on ImageNet-R dropped from $45.2\%$ when

using ImageNet-A for disagreement to $41.8\%$ when using ImageNet-R for disagreement. OOD detection performance also does not change much for any combination of regularizer and dataset used for disagreement with the biggest difference in detection AUROC scores 0.973 for Div regularizer and and ImageNet-A disagreement dataset vs 0.969 for Div regularizer and ImageNet-R disagreement dataset.

| Method | OOD | Ensemble Acc. | | | | | AUROC | | | |
|---|---|---|---|---|---|---|---|---|---|---|
| | | Val | IN-A | IN-R | C-1 | C-5 | C-1 | C-5 | iNat | OI |
| A2D | IN-A | **85.1** | **37.8** | **45.2** | **77.2** | **40.3** | 0.599 | **0.850** | 0.971 | 0.936 |
| A2D | IN-R | **85.1** | **37.8** | **45.2** | **77.2** | **40.3** | 0.599 | **0.850** | 0.971 | **0.939** |
| Div | IN-A | **85.1** | **37.8** | **45.2** | **77.2** | **40.3** | 0.599 | **0.850** | **0.973** | 0.937 |
| Div | IN-R | **85.1** | 35.7 | 41.8 | **77.2** | 40.2 | **0.600** | **0.850** | 0.969 | 0.938 |

Table 9: OOD Datasets for disagreement.

## A.12 Variations of the Stochastic Sum Size

We performed an additional evaluation (Table 10) that shows the benefit of controlling the stochastic sum size ($|\mathcal{I}|$) on the speed of training an ensemble. For example, to train an ensemble of size 5, the time required for 1 epoch grows from 53s for $|\mathcal{I}| = 2$ to 585s for $|\mathcal{I}| = 5$ (without stochastic sum). We could not train an ensemble of 50 models without stochastic sum with our resources, but it already requires 7244s for $|\mathcal{I}| = 10$ vs 2189s for $|\mathcal{I}| = 2$. Standard deviations of training epoch times are computed across 10 different epochs. The speed up is especially important for training an ensemble with 50 models. Since the number of model pairs grows from 1 to $C_{50}^2 = 1125$ in that case, a theoretical time for 1 epoch would be approximately 1225 times higher than the training time for $|\mathcal{I}| = 2$, i.e $\approx 0.5 \cdot 1125 = 663$ GPU hours. An important note here is that training time is affected by moving data between CPU and GPU (only the models used for loss computation are loaded to GPU in our implementation), therefore, it is hard to accurately predict epoch times.

| M | I | Epoch, s | Ensemble Acc. | | | | | AUROC | | | |
|---|---|---|---|---|---|---|---|---|---|---|---|
| | | | Val | IN-A | IN-R | C-1 | C-5 | C-1 | C-5 | iNat | OI |
| 5 | 2 | **53 ± 5** | **85.3** | **42.4** | **48.1** | **77.3** | **40.6** | 0.686 | 0.896 | **0.977** | **0.941** |
| 5 | 3 | 388 ± 28 | 85.2 | 41.4 | 47.4 | 77.2 | 40.5 | 0.682 | 0.892 | 0.975 | 0.939 |
| 5 | 4 | 423 ± 3 | 85.2 | 40.3 | 46.8 | 77.1 | 40.4 | 0.703 | 0.898 | 0.973 | 0.940 |
| 5 | 5 | 585 ± 111 | 85.1 | 37.6 | 44.9 | 77.0 | 40.2 | **0.711** | **0.903** | 0.970 | 0.937 |
| 50 | 2 | **2189 ± 86** | **83.7** | **50.1** | **54.0** | **75.9** | **39.4** | **0.600** | 0.824 | 0.934 | 0.878 |
| 50 | 5 | 4213 ± 5 | 83.6 | 49.2 | 53.4 | 75.8 | 39.2 | 0.598 | 0.827 | 0.942 | 0.892 |
| 50 | 10 | 7244 ± 27 | 83.4 | 48.5 | 53.0 | 75.6 | 39.1 | 0.597 | **0.828** | **0.945** | **0.896** |

Table 10: Variations of the stochastic sum size.

## A.13 Justification for stochastic sum

In this subsection, we will justify the stochastic sum described in § A.5.1 by showing that it provides an unbiased estimate of the total loss gradient. A simplified version of equation 6 for one batch of data can be written as follows:

$$\mathcal{L} = \mathcal{L}_{main} + \mathcal{L}_{div} = \frac{1}{|M|} \sum_{m \in M} \mathcal{L}(m) + \frac{1}{|P_M|} \sum_{p \in P_M} \mathcal{G}(p),$$

where $M$ is a set of all models in the ensemble; $\mathcal{L}(m)$ is loss computed for the $m$-th model output; p is a pair of models; $P_M$ is a set of all possible pairs of models from the set $M$; $\mathcal{G}(p)$ is a regularizer computed for the outputs of p.

Let's assume that on the current iteration we sampled a subset of models $I \subseteq M$. Then stochastic approximation of $\nabla L_{main}$ is:

$$\overline{\nabla\mathcal{L}_{main}} = \frac{1}{|I|} \sum_{m \in I} \nabla\mathcal{L}(m)$$

It is an unbiased estimate of the full gradient (by using the definition of expectation, its linearity property, and assuming a uniform probability distribution over models):

$$\mathbb{E}_{m \in M}[\overline{\nabla\mathcal{L}_{main}}] = \frac{1}{|I|} \sum_{m \in I} \mathbb{E}_{m \in M}[\nabla\mathcal{L}(m)] = \frac{1}{|I|} \cdot |I| \frac{1}{|M|} \sum_{m \in M} \nabla\mathcal{L}(m) = \nabla[\frac{1}{|M|} \sum_{m \in M} \mathcal{L}(m)] = \nabla\mathcal{L}_{main}$$

Similarly, stochastic approximation of $\nabla\mathcal{L}_{div}$ is:

$$\overline{\nabla\mathcal{L}_{div}} = \frac{1}{|P_I|} \sum_{p \in P_I} \nabla\mathcal{G}(p)$$

Which is also unbiased (by using the definition of expectation, its linearity property, and assuming a uniform probability distribution over pairs of models):

$$\mathbb{E}_{p \in P_M}[\overline{\nabla\mathcal{L}_{div}}] = \frac{1}{|P_I|} \sum_{p \in P_I} \mathbb{E}_{p \in P_M}[\nabla\mathcal{G}(p)] = \frac{1}{|P_I|} \cdot |P_I| \cdot \frac{1}{|P_M|} \sum_{p \in P_M} \nabla\mathcal{G}(p) = \nabla[\frac{1}{|P_M|} \sum_{p \in P_M} \mathcal{G}(p)] = \nabla L_{div},$$

where $P_I$ is a set of all pairs of models from $I$.

### A.14 JUSTIFICATION FOR $\alpha_n$

In this section we take a deeper look into gradients of the $\mathcal{L}_{\text{LDR}}$ (Equation 6) and justify why regularization weight $\alpha_n$ should depend on the sample-wise cross-entropy loss and scaled down by squared average cross-entropy loss in batch.

For the simplicity, we assume that ensemble contains only two models. For some fixed input $x$ with ground truth label $y$ we denote output probabilities as $f = p^1(x)$ and $g = p^2(x)$ for the first and second model correspondingly, while denoting their predictions as $\hat{f} = \arg\max_k f_k$ and $\hat{g} = \arg\max_k g_k$. We also omit the subscript of $\alpha_n$ for brevity and simply use $\alpha$ instead. In this case, the total training loss on a sample $(x, y)$ has the form:

$$\mathcal{L}_{\text{LDR}}(x, y) := -\log f_y - \log g_y - \alpha \log(f_{\hat{f}}(1 - g_{\hat{f}}) + g_{\hat{f}}(1 - f_{\hat{f}})) - \alpha \log(g_{\hat{g}}(1 - f_{\hat{g}}) + f_{\hat{g}}(1 - g_{\hat{g}}))$$

$$(17)$$

We can compute its partial derivatives (gradients) w.r.t. $f_y, f_{\hat{f}}, f_{\hat{g}}$ - probabilities predicted by model $f$ for the ground truth, for the prediction of model $f$ and for the prediction of model $g$ correspondingly:

$$\nabla_{f_y}\mathcal{L}_{\text{LDR}}(x, y) = -\frac{1}{f_y} \tag{18}$$

$$\nabla_{f_{\hat{f}}}\mathcal{L}_{\text{LDR}}(x, y) = \frac{\alpha(2g_{\hat{f}} - 1)}{f_{\hat{f}} + g_{\hat{f}} - 2f_{\hat{f}}g_{\hat{f}}} \tag{19}$$

$$\nabla_{f_{\hat{g}}}\mathcal{L}_{\text{LDR}}(x, y) = \frac{\alpha(2g_{\hat{g}} - 1)}{f_{\hat{g}} + g_{\hat{g}} - 2f_{\hat{g}}g_{\hat{g}}} \tag{20}$$

Similar partial derivatives can be obtained w.r.t. $g_y, g_{\hat{f}}, g_{\hat{g}}$.

Since $0 \leq C(f,g) = f_y + g_y - 2f_y g_y \leq 1$ for $0 \leq f_y \leq 1, 0 \leq g_y \leq 1$ (by examining critical and border points we can see that its minimum value is 0 and maximum value is 1 on this square), when $y \neq \hat{f} \neq \hat{g}$, the sign of the derivatives above depends only on the outputs of a single model (either $f$ or $g$). However, when $y = \hat{f}$ or $y = \hat{g}$ gradients start to clash with each other, i.e. the total gradient is obtained by summing two gradients with possibly opposite signs.

Let's consider the case $y = \hat{f}$:

$$\nabla_{f_y} \mathcal{L}_{\text{LDR}}(x,y) = -\frac{1}{f_y} + \frac{\alpha(2g_y - 1)}{C(f,g)} \tag{21}$$

It has two terms: the first term, $\dfrac{1}{f_y}$ has constant sign, while the sign of the second term, $\dfrac{\alpha(2g_y - 1)}{C(f,g)}$ depends on the value of $g_y$. This might lead to instabilities in training because the sign of the total gradient can flip during training depending on the current value of $g_y$.

To avoid such instabilities in gradient sign, we make weight $\alpha$ adaptive to the type of sample on which gradient is computed. For easy samples on which model makes correct predictions, i.e. high $f_y$, near-zero gradient value is desirable because we want to keep the prediction for such samples correct. For high loss samples, i.e. with low $f_y$, we want the gradient to be dominated by the second term that is responsible for models disagreement. Therefore, we make $\alpha$ inversely proportional to $f_y$ (to be precise we make it proportional to $-\log f_y$ for computational stability reasons).

The need for inverse proportion can be seen after checking when absolute values of the two gradient terms equal to each other:

$$\frac{1}{f_y} = \frac{|-1|}{|f_y|} = \frac{\alpha|2g_y - 1|}{C(f,g)} \tag{22}$$

$$f_y = \frac{C(f,g)}{\alpha|2g_y - 1|} \tag{23}$$

If we set $\alpha$ to some constant value $\overline{\alpha}$, there will always be a value of $\overline{f_y} = \dfrac{C(f_y, g_y)}{\overline{\alpha}|2g_y - 1|}$, such that for $f_y(x) < \overline{f_y}$ the first term dominates the gradient and for $f_y > \overline{f_y(x)}$ the second term dominates the gradient. Such behavior will again lead to clashes between the terms depending on the value of $f_y$.

The only way to avoid such clashes is to set $\alpha$ proportional to $\dfrac{1}{f_y}$, i.e. $\alpha = \gamma \dfrac{1}{f_y}$, for some $\gamma > 0$. Then from Equation 22 we will get the following condition for the second term dominance in the total gradient (for $0 \leq C(f,g) \leq 1$):

$$\frac{1}{f_y} \leq \frac{\gamma|2g_y - 1|}{f_y C(f,g)} \tag{24}$$

$$\gamma^{-1} \leq |2g_y - 1| \leq \frac{|2g_y - 1|}{C(f,g)} \tag{25}$$

However, making $\alpha$ inversely proportional to $f_y$ is not enough, as in the beginning of the training when $f_y$ is small on all training samples, the second term always dominates the gradient in Equation 21 resulting only in outputs diversification and neglecting the classification task. To solve this problem, we scale down $\alpha$ by a squared average cross-entropy loss in batch as shown in Equation 5, the square

is important to keep the average value of $\alpha$ dependent on the average cross entropy as explained in Section 3.3.

Similar reasoning can be applied to the cases, when $y = \hat{g}$ or $y = \hat{f} = \hat{g}$. The argument holds for gradients computed w.r.t. $g_y, g_{\hat{f}}, g_{\hat{g}}$ and scenarios with more than two models in the ensemble.

