# OpenReview forum: "Rethinking OOD Detection at Scale through Ensemble Diversity"
_ICLR.cc/2026/Conference — ICLR 2026 Conference Withdrawn Submission_

### Official Review · Reviewer_8YYZ · 2025-10-18

**Soundness:** 2
**Presentation:** 3
**Contribution:** 3
**Rating:** 4
**Confidence:** 4

**Summary:**

This work aims at enhancing the ensemble diversity without reliance on external out-of-distribution (OOD) data in OOD detection. A key premise of this work is that the large-scale training data has contained both the high-density IND (in-distribution) data and the low-density OOD data. This work explores how to identify such OOD data from the training dataset according to the training loss: disagreement on high-loss training samples can somehow replace disagreement on ood data. This insight motivates the proposed regularizer named Loss-guided Diversification Regularizer (LDR). LDR designs a sample-wise loss coefficient by leveraging the training data only, and is compatible with the diversity loss of existing ensemble training methods. An associated detection score named Predictive Diversity Score (PDS) is further proposed. Theories and experiments are provided in this work.

**Strengths:**

1.	Exploring diversities among multiple models without accessing additional OOD data is appreciated, which is an important and practical topic.
2.	Both theories and empirical results are provided to validate the effectiveness of the proposed training regularization term and the OOD detection metric.
3.	The writing is good and easy to follow.

**Weaknesses:**

**Major weaknesses**

1.	About the high-density and low-density samples

A key premise of this work is that large-scale datasets contains both high-density and low-density samples, where the latter can be viewed as OOD data so as to avoid external OOD data in training. While this premise sounds reasonable, there are no intuitive numerical results or visual illustrations in the manuscript to support it. This premise is directly formularized as an assumption in theories. Therefore, it would strengthen this work by providing numerical analyzes or illustrations to validate the presence of low-density samples in practical large-scale datasets such as ImageNet-1K.

2.	A further discussion

This work addresses the ensemble diversity from the perspective of training data. In contrast, another related work [a] explores the ensemble diversity from the loss landscape perspective. That is, it is highlighted in [a] that independently-trained models inherently show diversity on OOD data, leading to a high variance among their detection results. The authors are suggested to give a discussion with [a] to enrich the literature review.

3.	Empirical comparisons on more OOD datasets

In experiments, the involved OOD datasets are Openimage-O, iNaturalist and ImageNet-C, which is slightly insufficient. It would be valuable to provide results on more OOD datasets following the commonly used settings in the field, such as SUN, Places, Textures and ImageNet-O.

[a] Revisiting Deep Ensemble for Out-of-Distribution Detection: A Loss Landscape Perspective. IJCV 2024.

**Minor weaknesses**

4.	Some mathematical notations lack explicit definitions. In line 136, the definition of $\rm Mix(\cdot,\cdot)$ is missing. In line 141, what does $\cal Y^X$ indicate?

5.	I think the comparisons between PDS and other ensemble detection scores in Table 6 are important and primary results, which should be put in main text, instead of the appendix.

**Questions:**

My questions correspond to the 5 weaknesses outlined above. I would like to increase my score should all these points be adequately addressed.

---

### Official Review · Reviewer_z47R · 2025-10-24

**Soundness:** 2
**Presentation:** 3
**Contribution:** 2
**Rating:** 2
**Confidence:** 4

**Summary:**

The paper proposes a unified pipeline that integrates label error detection [1] and ensemble diversification [2, 3, 7], leveraging automatically identified OOD-like data [4, 5, 6].

**Strengths:**

- The approach extracts OOD-like samples dynamically (“on-the-fly”) from a large, noisy training set, removing the need for explicit OOD data.
- The proposed Predictive Diversity Score (PDS) provides a novel uncertainty metric based on ensemble disagreement.
- The paper is clearly written and evaluates on large-scale datasets such as ImageNet, iNaturalist, and OpenImages.
- The paper demonstrates improved performance in a wide range of OOD detection tasks.

**Weaknesses:**

- The paper redefines OOD as “low-density training data”, which diverges from the standard definition (data from a different distribution). This conceptual shift leads to inconsistencies with prior theory and practice, weakening the theoretical grounding of the method.
- The theoretical section contains unclear or flawed derivations. For instance:
  - The auxiliary distribution Q is not properly introduced.
  - In some edge cases (e.g. Q = P), the theoretical claims in Theorem 1 become contradictory.
  - Theorem 1 assumes a finite number of models M > 1, yet the proof relies on an infinite ensemble.
- There is no direct comparison with a simple baseline combining existing label error detection [1] and ensemble diversification methods [5, 6].
- The use of a “shallow” Deep Ensemble (training only the last two layers) raises concerns, as Deep Ensembles are strong baselines. The justification (matching computational complexity) is insufficient, and there is no study of the trade-off between efficiency, convergence speed, and performance.
- Experimental robustness is not demonstrated. Standard deviations over multiple random seeds (initializations) are missing, making it difficult to assess the stability of results.
- Several highly relevant baselines are omitted:
  - Ensemble diversification methods without external data [2, 3, 7].
  - Modern OOD scoring functions (e.g., logits-based approaches) [3, 8].

**Questions:**

- How can the proposed definition of OOD (as low-density regions within the training set) be rigorously distinguished from long-tail or rare in-distribution samples?
- In what ways does the proposed pipeline differ from a simple composition of label error detection and existing ensemble diversification methods?
- Are there existing works that adopt a similar evaluation setup (datasets, architectures, and metrics)? Why not evaluate on a standardized benchmark such as OpenOOD v1.5 [9]?

[1] Model-Agnostic Label Quality Scoring to Detect Real-World Label Errors (2022)

[2] DICE: Diversity in Deep Ensembles via Conditional Redundancy Adversarial Estimation (2021)

[3] Diversifying Deep Ensembles: A Saliency Map Approach for Enhanced OOD Detection, Calibration, and Accuracy (2024)

[4] Deep anomaly detection with outlier exposure (2019)

[5] Diversify and disambiguate: Out-of-distribution robustness via disagreement (2023)

[6] Agree to disagree: Diversity through disagreement for better transferability (2023)

[7] Improving adversarial robustness via promoting ensemble diversity (2019)

[8] Scaling out-of-distribution detection for real-world settings (2022)

[9] Openood v1.5: Enhanced benchmark for out-of-distribution detection (2023)

---

### Official Review · Reviewer_pWP7 · 2025-10-31

**Soundness:** 3
**Presentation:** 2
**Contribution:** 2
**Rating:** 4
**Confidence:** 3

**Summary:**

This paper proposes a novel method called LDR (Loss-guided Diversification Regularizer) for training Diverse Deep Ensembles in large-scale scenarios, aiming to improve the performance of Out-of-Distribution (OOD) Detection. Its core idea is to dynamically encourage disagreement among models on high-loss samples during the training process, thereby enhancing the models' ability to estimate the uncertainty of OOD samples.

**Strengths:**

1, The motivation is practical which identifies the challenges faced by existing OOD detection methods in large-scale, real-world scenarios, and attributes the problem to insufficient model diversity.
2. The design of LDR is highly straightforward. It connects the disagreement regularization term with sample loss through an adaptive weight, avoiding explicit reliance on OOD data. This is a significant advantage in large-scale applications.
3. A new uncertainty metric PDS (Predictive Diversity Score) is introduced. It reflects the diversity of the ensemble more directly than the traditional BMA (Bayesian Model Averaging) entropy, making it a valuable contribution.

**Weaknesses:**

1. The boundary between the core idea of this paper and existing works (A2D, DivDis) is not sufficiently clear. The paper needs to more explicitly explain the fundamental differences and incremental contributions of LDR compared to A2D and DivDis.

2. The baselines used for comparison appear outdated. The paper should include additional experiments comparing against the latest state-of-the-art (SOTA) OOD detection methods.

3. The analysis of computational overhead is inadequate. Although LDR significantly reduces training cost via stochastic summation, its inference cost requiring the execution of M models，which is M times higher than that of a single model approach. The paper fails to discuss this trade-off, which is critical for real-world deployment.

**Questions:**

Please refer  to the Weaknesses.

---

### Official Review · Reviewer_2m72 · 2025-11-01

**Soundness:** 3
**Presentation:** 3
**Contribution:** 2
**Rating:** 4
**Confidence:** 4

**Summary:**

The paper argues that treating the whole training set as purely in-distribution is wrong at web/Imagenet scale because the data already contains low-density, OOD-like examples. It proposes a loss-guided diversification regularizer (LDR) that, during ensemble training, finds high–cross-entropy samples inside the training set and forces different ensemble members to disagree on them, while using a stochastic pairing trick to keep the cost constant in the ensemble size. It also introduces a predictive diversity score (PDS) that directly measures ensemble prediction spread and turns that into an OOD uncertainty signal.

**Strengths:**

I like the idea of this paper. The paper clearly identifies a real mismatch between the classic OOD setup (clean ID + separate OOD) and modern, messy, web-sourced datasets, arguing—very concretely—that ImageNet itself already contains low-density, OOD-like, or mislabeled points. That’s a simple but nontrivial reframing of the task, and it is well motivated in the intro with citations to long-tail / noisy-ImageNet work.

**Weaknesses:**

1. The whole method rests on the claim that ImageNet-scale data is actually a mixture Mix(P,Q), so high-loss points you see during training are in fact “low-density / OOD-like” examples. But the paper never quantitatively validates that the selected high-loss samples are (i) truly off-manifold, (ii) mislabeled, or (iii) distributionally closer to the test-time OOD sets (iNat, OI, IN-C) than to ID.

2. High loss can come from label noise, hard positives, fine-grained classes, class imbalance, spurious backgrounds, or plain optimization lag—none of these is necessarily OOD. Encouraging disagreement exactly on such data may amplify noise or minority-class instability. The paper gestures at label noise in passing but does not run any noise-robustness or long-tailed/stress tests to show LDR isn’t just learning to disagree on bad labels.

3. The disagreement-based characterization (Theorem 1) assumes access to an ensemble of solutions with zero error on
P; the main theorem on high-loss sets (Theorem 2) needs lower-bounded disagreement on a set
H. But in practice they train shallow ImageNet ensembles, partially frozen, with nonzero training error and with mixed
P/Q. The paper doesn’t show how tight these assumptions are in practice, or how sensitive LDR is when the assumptions fail.

4. All experiments are in vision, on ImageNet-val as ID, and on a few standard OOD sets (iNat, OI, IN-C). But the abstract/intro talk about “large-scale applications” and “web-sourced heterogeneity.” Furthermore, the proposed method has not been evaluated on a standard benchmark (CIFAR benchmark). Although the CIFAR benchmark is smaller, it is indeed a more challenging benchmark. If your labeled set is CIFAR-100-size, LDR may not find the “OOD-like tail” it relies on. The paper should run a low-data / medium-data experiment to show what happens when the “large-scale assumption” is violated—and maybe combine LDR with pseudo-labeling on the unlabeled pool. Right now the method is “scalable” only in the sense that it needs scale.

5. Impoartant Related works not discussed: There havebeen lots of research works on ensemble-based OOD detection, but none of the related works is discussed. [1-5]

[1] Fang K, Tao Q, Huang X, et al. Revisiting deep ensemble for out-of-distribution detection: A loss landscape perspective. IJCV, 2024

[2] Vyas A, Jammalamadaka N, Zhu X, et al. Out-of-distribution detection using an ensemble of self supervised leave-out classifiers. ECCV 2018

[3] Xu C, Yu F, Xu Z, et al. Out-of-distribution detection via deep multi-comprehension ensemble. ICML 2024.

[4] Ren J, Liu P J, Fertig E, et al. Likelihood ratios for out-of-distribution detection, NeurIPS 2019.

[5] Fort S, Ren J, Lakshminarayanan B. Exploring the limits of out-of-distribution detection, NeurIPS 2021.

**Questions:**

1. Why is “high loss ⇒ OOD-like” the right proxy? Did you try alternative difficulty signals—e.g., low-confidence but consistent predictions, large feature distance to class prototypes, or disagreement without high loss—and if so, how did they compare to pure CE-based mining? This would help separate the effect of “mining anything hard” from your specific loss-guided design.

2. How robust is LDR if a nontrivial portion of the high-loss set is in fact noisy labels or minority classes? Right now LDR explicitly pushes models to disagree on high-loss points; if many of those are rare/long-tailed classes, that could actually hurt representation of those classes. Can you run (or at least discuss) a setting with synthetic label noise or an ImageNet-LT–style split to show that LDR isn’t just amplifying noise?

---

### Official Review · Reviewer_6LcJ · 2025-11-03

**Soundness:** 2
**Presentation:** 3
**Contribution:** 2
**Rating:** 4
**Confidence:** 4

**Summary:**

This paper challenges the traditional view of out-of-distribution (OOD) detection by arguing that large-scale training sets are not purely in-distribution (ID) and inherently contain the low-density, "OOD-like" samples needed for robust ensemble training. It introduces the Loss-guided Diversification Regulariser (LDR), a method that eliminates the need for external OOD data by identifying these internal samples based on their high cross-entropy loss. LDR trains an ensemble to specifically disagree on these high-loss samples while using a stochastic pairing strategy to reduce computational complexity from quadratic to constant, ensuring scalability.

**Strengths:**

1.	Novel Conceptual Reframing: The paper claims a core assumption in OOD detection. It moves beyond the "training set = ID" oversimplification and leverages the insight that large-scale datasets already contain "OOD-like" samples.
2.	It proposes a simple yet effective proxy for identifying these "OOD-like" internal samples: high cross-entropy loss. The paper provides a theoretical justification (Theorem 2) that forcing disagreement on these high-loss samples increases disagreement on actual OOD data.

**Weaknesses:**

1.	The paper's central premise is that high-loss samples within the training set are a valid substitute for an external OOD dataset to train ensemble disagreement. This is a powerful idea, but its validity is insufficiently explored. The high-loss group is heterogeneous. It likely contains not only the "ambiguous" or "OOD-like" samples the authors are targeting but also (a) hard in-distribution (ID) samples and (b) mislabeled samples. The LDR regularizer forces disagreement on all of them. Forcing disagreement on hard-ID samples could hurt generalization on the ID task itself, while forcing disagreement on mislabeled data is conceptually problematic and may not lead to the desired "epistemic uncertainty" signal. The paper lacks a rigorous analysis to disentangle these cases.
2.	The method's effectiveness is not tested in controlled settings where the training data is small or guaranteed to lack OOD samples. Its premise relies on large, heterogeneous datasets containing high-loss "OOD-like" samples.
3.	A key selling point is “O(1) w.r.t. M thanks to stochastic pairing.” But we don’t see actual numbers: what’s the per-step overhead vs a plain 5× shallow ensemble? What batch sizes were needed to make the high-loss weighting stable? How many SGD steps until the high-loss set stops changing? Without real costs, it’s hard for practitioners to decide if LDR is better than “just train 2 more seeds” or “diverse HPs,” which in Table 2 already gets good AUROC. Add a small “cost vs AUROC” plot to make the practical benefit concrete.

**Questions:**

Could you provide a more detailed analysis (perhaps qualitative or quantitative) of the composition of the samples LDR selects (i.e., those with high αn weights)?
How does the method ensure that forcing disagreement on "hard-ID" samples does not harm in-distribution performance, or that forcing disagreement on mislabeled data leads to the desired epistemic uncertainty signal?
This suggests the O(1) approximation comes at a cost to performance. Could you please analyze this trade-off between computational efficiency and detection performance more thoroughly?
How much performance is being left on the table by this approximation, especially for large ensembles like M=50?

---

### Note · Authors · 2025-11-12

I have read and agree with the venue's withdrawal policy on behalf of myself and my co-authors.